

# Joint probability analysis of storm surge and wave caused by tropical cyclone for the estimation of protection standard: a case study on the eastern coast of the Leizhou Peninsula and Hainan Island of China

Zhang Haixia[1,2,3,4], Cheng Meng[1,2,3,4], Fang Weihua[1,2,3,4]

[1] Key Laboratory of Environmental Change and Natural Disasters, Ministry of Education, Beijing Normal University, 100875, Beijing, China

[2] Academy of Disaster Risk Science, Faculty of Geographical Science, Beijing Normal University, 100875, Beijing, China

[3] Southern Marine Science and Engineering Guangdong Laboratory (Guangzhou), 511458, Guangzhou, Guangdong, China

[4] State Key Laboratory of Earth Surface Processes and Resource Ecology (ESPRE), Beijing Normal University, 100875,
Beijing, China

*Correspondence to*: Weihua Fang (weihua.fang@bnu.edu.cn)

**Abstract.** Quantitatively estimating combined storm surge and wave hazards provides scientific guidance for disaster prevention, mitigation, and relief in coastal cities. The marginal and copula functions are preferred based on the Kolmogorov–Smirnov test pass rate, and the relationship between storm surge and wave is quantitatively evaluated using the optimal function,
then the bivariate risk probabilities are estimated. The results show that the generalized extreme value function and Gumbel copula function are suitable for fitting the marginal and joint distribution characteristics of the surge height and significant wave height in this study area, respectively. Second, the surge height shows an increasing trend closer to the coastline, and the significant wave height is higher further from the coastline. Third, when one variable is constant, the simultaneous, joint, and conditional risk probability tends to decrease as the other variable increases. In actual engineering design, improving the
protection standard can effectively reduce the bivariate risk probability. In addition, we can estimate the optimal design criteria for different joint return periods by the constraint condition and objective functions. This study shows that the bivariate copula function can effectively evaluate the risk probability for different scenarios, which provides a reference for optimizing engineering protection standards.

**Keywords:** Joint probability analysis, Storm surge and wave, Copula function, Tropical cyclone, Leizhou Peninsula and
Hainan Island



## 1 Introduction

In the background of rising sea surface temperatures (SSTs), the lifetime of tropical cyclones (TCs) is extended and their intensity is enhanced. In particular, the frequency of severe tropical storms (STSs) and storms with higher intensity levels has significantly increased (Emanuel, 2005; Elsner et al., 2008; Sun et al., 2017), triggering more tremendous storm surges and
wave disasters and causing severe casualties and property damage in offshore and coastal areas (Chen and Yu, 2017). Accurate and rapid assessment of tropical cyclone storm surge and wave occurrence probability and intensity is essential for actively and effectively preventing extreme disaster risks (Zhang and Wang, 2021; Teena et al., 2012).

Tropical cyclone storm surges and waves significantly threaten marine fishing, aquaculture, marine transportation, offshore oil and gas development, and other offshore production activities (Jin et al., 2018; Guo et al., 2020). When storm surges are
accompanied by waves near the coast, they cause severe damage to coastal dikes, breakwaters, revetments, industrial facilities, roads, and bridges (Rao et al., 2012; Hughes and Nadal, 2009). When the sea level in front of the sea dike caused by a superstorm surge exceeds the top of the sea dike or when gales and waves cause damage to the wave wall at the top of the sea dike, it can cause the sea dike to suffer from both overtopping waves and overflowing surges (Pan et al., 2013, 2019; Li et al., 2012). When overtopping and overflowing occur, combined with the effect of sea flooding, they can inundate coastal towns,
farmland, salt flats, and aquaculture areas, which exacerbates the impact of the disaster (Jones et al., 2013; Xian et al., 2015; Shi et al., 2022; Niedoroda et al., 2010). According to statistics, the tropical cyclone *Dove* (ID: 1713) storm surge and wave disaster in Guangdong Province caused the following impacts: 6 deaths (including those missing); direct economic losses of 5.154 billion yuan; affected population of 1,128,600 people; collapse of 25 houses; affected aquaculture area of $1.824 \times 10^4$ hm$^2$; loss of 330 farming equipment facilities; loss of 43 fishing boats; and damage to 1 fishing harbor, to 1.22 km of docks,
to 240.18 km of breakwater, to sea dikes, to 532.08 km of revetments, to 0.02 km of roads, and to a farmland inundation area of $1.374 \times 10^4$ hm$^2$ (SOA, 2018).

Research on the construction of a single-causing factor assessment index system, disaster threshold determination, and hazard assessment of natural hazards are relatively mature, and numerous studies on the risk of the single-causing factor of tropical cyclone storm surges and sea waves have been conducted by combining statistical analysis and numerical simulation (Lin et
al., 2010; Shi et al., 2020; Teena et al., 2012). Assume that the number of historical observation samples is insufficient. In this

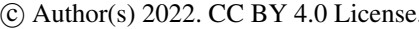



case, a tropical cyclone can be synthesized by the joint probability method optimal sampling (JPM-OS), or a tropical cyclone stochastic event set can be constructed by the Monte Carlo method. An extensive collection of storm surge events can be simulated using the tropical cyclone parameter model and hydrodynamic model SLOSH with ADCIRC. On this basis, a statistical analysis such as the extreme value function estimates the probability density distribution and return period of surge

height (Niedoroda et al., 2010). However, tropical cyclone disasters in coastal areas are caused by the combined effect of storm surges, waves, and other disaster-causing factors, and the mechanism of disaster generation is very complex. The interaction, concurrency, or chain of disaster-causing factors makes the comprehensive hazard assessment of compound hazards with significant uncertainties. Therefore, the study of the joint probability distribution of tropical cyclone storm surges and waves is conducive to improving the accuracy and precision of joint hazard assessment, thus laying the foundation for formulating

effective response measures and disaster prevention and mitigation policies (Xu et al., 2022). Presently, a few scholars have conducted studies on the interaction and joint distribution of storm surges and waves (Chen et al., 2019; Li et al., 2021), but they focused on the simulation studies of coupled hydrodynamic models of storm surges and waves (Bastidas et al., 2016; Chen and Yu, 2017).

The copula function is widely employed in studies of the joint distribution characteristics of multidimensional, natural disaster-

causing factors because it does not restrict the marginal distribution function, is easily extended to multiple dimensions, and can be flexibly constructed to a multidimensional joint distribution. Studies have analyzed the correlation of multicausal factors, such as tropical cyclone gale–rainstorms (Hou et al., 2019; Ye and Fang, 2018; Dong et al., 2017), rainstorm-storm surges (Xu et al., 2022), gale-storm surges (Trepanier et al., 2015), and storm surge waves (Corbella and Stretch, 2013; Wahl et al., 2012). However, the constructed joint distribution is still a probabilistic result, and further search for constraint relations is needed to

provide a basis and guidance for disaster prevention and mitigation. Therefore, this paper quantitatively analyzes the occurrence probability of storm surge and wave combinations based on the fitting results of the copula function, and on this basis, explores the influence of the change in the design criteria of surge height and significant wave height on the bivariate joint probability so that it can be applied to the design of engineering fortification.

In the design process of sea dikes, breakwaters, and harbors, the surge height and significant wave height in different return

periods is separately considered (MWR, 2014; MOT, 2015, 2018), disregarding the correlation between storm surge and waves

so that the calculated water level may be underestimated or overestimated. Therefore, this paper explores the joint probability distribution characteristics of storm surge-wave composite hazards, calculates storm surges and waves for different joint return periods, and provides a reference basis for the design criteria of engineering facilities.

In this paper, we use the tropical cyclone surge height and significant wave height obtained from numerical simulation, select the region east of the Leizhou Peninsula and Hainan Island as the study area, optimize the marginal distribution function and copula function by the passing rate of the K-S test, and use the maximum likelihood method to estimate the function parameters. Second, the correlation between surge height and significant wave height is quantified using the optimal function to calculate the simultaneous risk probability, joint risk probability, conditional risk probability, and risk probability of the hazard factors at different combinations of levels. Last, the change values of bivariate risk probability are quantitatively evaluated for engineering protection standards after increasing the surge height or significant wave height design standards. The optimal values of surge height and significant wave height for different joint return periods are estimated. This study provides methodological support and a theoretical reference for the joint probabilistic characterization of tropical cyclone hazards and the design of engineering protection criteria.

## 2 Study area and data

### 2.1 Study area

Based on the location of the nodes of the triangular network in the storm surge (Section 2.3) and wave datasets (Section 2.4), we select the region with a dense distribution of both as the study area, and the finalized spatial range is 110°E - 113°E, 18°N - 22°N (Figure 1b). This area is east of the Leizhou Peninsula and Hainan Island in the South China Sea, which is also the area most frequently affected by tropical cyclones in China. Based on the dataset of surge height (SH) and significant wave height (SWH) of tropical cyclones, we screen 86 historical tropical cyclone (TC) events that simultaneously affected the study area from 1949 to 2013 for joint probability characteristics analysis of storm surge and wave.

### 2.2 Best tracks of TCs

The best track dataset of historical TCs in the Northwest Pacific (NWP) is obtained from the Tropical Cyclone Data Center of





the China Meteorological Administration (CMA). The CMA records in detail the location (longitude and latitude), time (year,

month, day, hour, UTC), central minimum pressure, and 2-minute average near-center maximum sustained wind speed (MSW)

for every 6-hour track point of each TC event since 1949 (Lu et al., 2021). The landfall of TCs in China is concentrated on the

southeast coast, especially in the coastal areas of the South China Sea. Figure 1a shows the spatial distribution of the best track

and maximum sustained wind speed of 86 historical TCs screened in this study from 1949 to 2013.

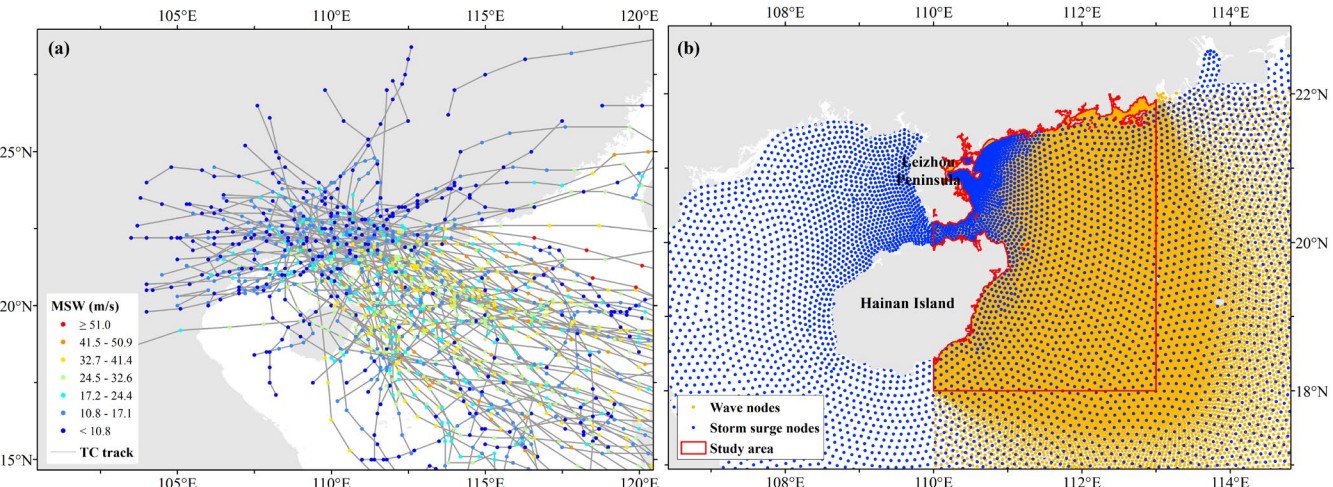

**Figure 1: Best track and MSW of 86 TCs in this study from 1949 to 2013 (a) and the study area for the joint probability analysis of**
**storm surges and waves of TCs (b).**

**2.3 Surge heights**

The TC surge heights (SHs) are derived from the Advanced Circulation Model simulation results (ADCIRC), which include

the SHs of 119 TCs affecting the Leizhou Peninsula from 1949 to 2013 (Liu et al., 2018; Li et al., 2016). The simulation results

are the total water level after the superposition of the water gain caused by a tropical cyclone and astronomical tide, and the

time step of the simulation is 30 minutes. To improve the simulation accuracy and computing speed of the hot spot area, the

model adopts a triangular network with nested small- and large-area grids and the resolutions of different area grids are set in

a gradual resolution range from 0.0039° to 0.3°. Comparing the simulation values with the measured surge height at the

observation sites, we discover that the absolute standard error is 47 cm, that the relative standard error is 22%, and that the

simulation results are similar to the observed values in most cases. Thus, the dataset could be used to assess the hazard of TC

storm surges. Figure 2a shows an example of the simulation results of the surge height of TC *Nasha* (ID:1117) at a specific



moment.

## 2.4 Significant wave heights

The TC significant wave heights (SWHs) are simulated using the SWAN model, including the SWHs of 102 TC events
affecting the Leizhou Peninsula from 1949 to 2014 (Li et al., 2016). The simulation results include indicators such as significant

wave height, mean period, and wave direction, and the simulation time step is 1 hour. The model also uses a triangular network

with nested small- and large-area grids and gradual resolution, but the nodes' scopes and locations differ from those of the

storm surge model. A comparison of the observed data of buoy stations with the simulated values reveals that the unstructured

grid can well reflect the wave variation conditions in the sea. In addition, the mean absolute error and root mean square error
of the simulated results of the locally encrypted unstructured triangular network are the smallest, indicating that the data can

effectively reproduce the wave distribution during tropical cyclones. In this paper, we mainly choose the SWH as an indicator

of tropical cyclone wave hazards. Figure 2b shows an example of the significant wave height of TC *Nasha* (ID: 1170) at a

specific moment.

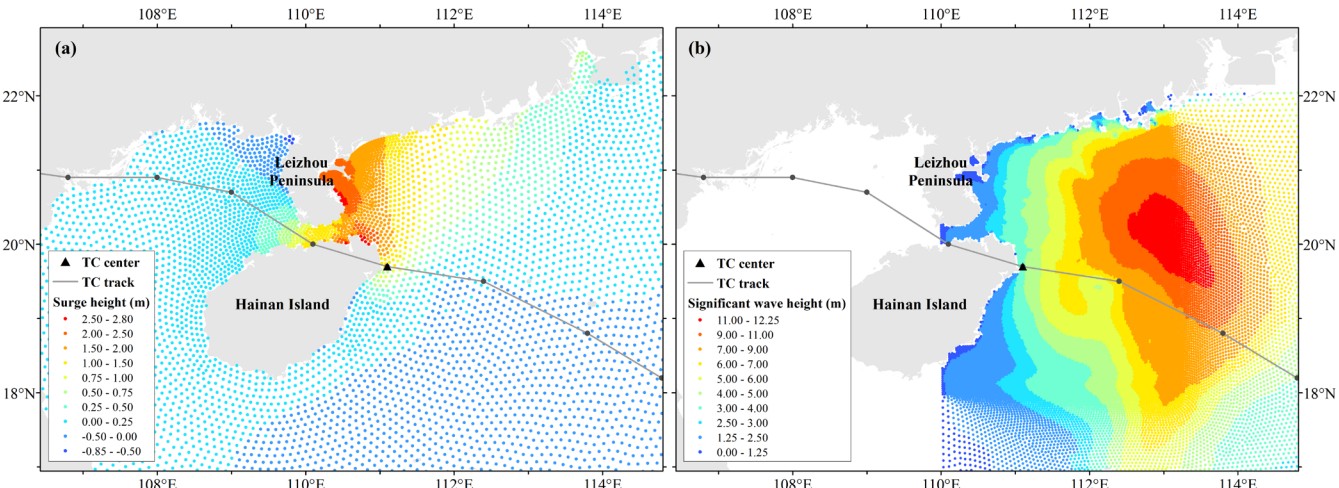

**Figure 2: Distribution of surge height (a) and significant wave height (b) at a specific moment of TC *Nasha* (ID: 1117) (UTC:
2011.9.29 6:00:00)**

## 3 Methods

Sklar (Sklar, 1973) proposed that any multivariate joint distribution can be composed of multiple univariate marginal





distribution functions and a copula function describing the correlation structure among various variables, i.e., Sklar's theorem,

which is the basis of copula theory and applications. Therefore, the copula function refers to the role that connects the marginal

distribution functions and is also known as the link function, which can accurately calculate the risk probability with the

combination of multivariate encounters according to the actual situation and is widely utilized in the joint probability analysis

of multihazard indicators in natural disasters.

$$F_{X,Y}(x,y) = C[F_X(x), F_Y(y)] \tag{1}$$

where $F_X(x)$ and $F_Y(y)$ are the probability distribution functions of variables $X$ and $Y$, respectively; $C$ is the copula

function; and $F_{X,Y}(x,y)$ is the joint distribution function of the two-dimensional variables $(X, Y)$ (Nelsen, 2006).

### 3.1 Marginal distribution function

The marginal distribution function fitting means that the probability density function (PDF) and cumulative distribution

function (CDF) of the univariate are constructed by intensity-frequency analysis to reflect the likelihood of the occurrence of

the univariate at different intensities. The method is widely utilized in natural hazard assessments such as tropical cyclones,

floods, droughts, and earthquakes. This paper selects five commonly employed marginal distribution functions for the annual

extreme values of tropical cyclone storm surges and waves, including the Gumbel, Weibull, gamma, exponential, and

generalized extreme value (GEV) distributions. Next, we use the maximum likelihood method to estimate the fitting parameters,

based on which we determine the goodness of fit of each node by the Kolmogorov–Smirnov (K-S) test. An optimal function

is selected as the univariate marginal distribution function for all nodes, and its PDF and CDF are fitted.

### 3.2 Bivariate copula function

There are three families of copula functions: elliptical, Archimedean, and quadratic. Archimedean copula functions are simple

to construct and contain only one parameter, which is convenient for solving and has been widely used in hydrological

multivariate frequency calculations. The commonly employed Archimedean copula includes Gumbel, Clayton, and Frank

(Table 1), which are chosen in this paper to analyze the joint probabilities of the marginal distributions of two variables, tropical

cyclone storm surge and waves, and to estimate the parameters of the copula function using the maximum likelihood method.



Next, we fit the goodness-of-fit of copula functions for the bivariate tropical cyclone storm surge and waves at each node by the K-S test. According to the passing rate of the K-S test at the sample points, an optimal function is selected as the copula function for all nodes of the two-dimensional variables, and the PDF and CDF of the two-dimensional variables are calculated.

**Table 1 Formulas and parameter ranges for three types of bivariate Archimedean copula functions.**

| Name of copula | Bivariate Copula | Parameter $\theta$ |
|---|---|---|
| Clayton | $C_\theta(u,v) = [max\{u^{-\theta} + v^{-\theta} - 1; 0\}]^{-1/\theta}$ | $\theta \in [-1, \infty)\setminus\{0\}$ |
| Frank | $C_\theta(u,v) = -\frac{1}{\theta} log \left[1 + \frac{(e^{-\theta u} - 1)(e^{-\theta v} - 1)}{e^{-\theta} - 1}\right]$ | $\theta \in R\setminus\{0\}$ |
| Gumbel | $C_\theta(u,v) = exp\left[-((-log(u))^\theta + (-log(v))^\theta)^{\frac{1}{\theta}}\right]$ | $\theta \in [1, \infty)$ |

### 3.3 Joint probability of storm surges and waves

#### 3.3.1 Univariate return period

The return period (RP) indicates the period of natural hazard events; it is a crucial indicator for quantifying the hazard level, which is widely utilized in disaster risk analysis. The formula for the return period of a single disaster-causing factor $RP_X$ is expressed as follows.

$$RP_X = \frac{E_L}{1 - F_X(x)} = \frac{E_L}{1 - P(X \leq x)} \tag{2}$$

where $F_X(x) = P(X \leq x)$ is the marginal distribution function of the univariate $X$, and $E_L$ denotes the time interval of the sample series of the univariate $X$. The value is taken as 1 in this paper.

#### 3.3.2 Bivariate risk probability and return period

Based on the copula function, it can quantitatively estimate the probability of multiple variables greater than a specified threshold. The bivariate risk probability refers to the likelihood that various conditions will simultaneously hold, and the bivariate return period refers to the average time interval required for multiple states to be simultaneously greater than a certain threshold.

The definitions of three types of joint risk probabilities and return periods are given according to the univariate return period





formula. The first type is when two variables simultaneously reach a given threshold, which will be defined as the simultaneous risk probability $P_\cap$ (Eq. 3) and simultaneous return period $RP_\cap$ (Eq. 4). The second type is that at least one variable reaches a given threshold, which is defined as the joint risk probability $P_\cup$ (Eq. 5) and joint return period $RP_\cup$ (Eq. 6). The third type is the conditional risk probability $P_|$ (Eq. 7) and conditional return period $RP_|$ (Eq. 8) When one of the variables reaches a given threshold, the other variable reaches a certain threshold. The calculation formula is expressed as follows:

$$P_\cap = P\big((X > x) \cap (Y > y)\big) = 1 - P(X \le x) - P(Y \le y) + P(X \le x, Y \le y)$$
$$= 1 - F_X(x) - F_Y(y) + F_{X,Y}(x,y) \tag{3}$$

$$RP_\cap = \frac{E_L}{P\big((X > x) \cap (Y > y)\big)} = \frac{E_L}{1 - F_X(x) - F_Y(y) + F_{X,Y}(x,y)} \tag{4}$$

$$P_\cup = P\big((X > x) \cup (Y > y)\big) = 1 - P(X \le x, Y \le y) = 1 - F_{X,Y}(x,y) \tag{5}$$

$$RP_\cup = \frac{E_L}{P\big((X > x) \cup (Y > y)\big)} = \frac{E_L}{1 - F_{X,Y}(x,y)} \tag{6}$$

$$P_| = P\big((X > x)|(Y > y)\big) = \frac{P(X > x, Y > y)}{P(Y > y)} = \frac{1 - P(X \le x) - P(Y \le y) + P(X \le x, Y \le y)}{1 - P(Y \le y)}$$
$$= \frac{1 - F_X(x) - F_Y(y) + F_{X,Y}(x,y)}{1 - F_Y(y)} \tag{7}$$

$$RP_| = \frac{E_L}{P\big((X > x)|(Y > y)\big)} = \frac{E_L \cdot \big(1 - F_Y(y)\big)}{1 - F_X(x) - F_Y(y) + F_{X,Y}(x,y)} \tag{8}$$

where $F_X(x)$ and $F_Y(y)$ are the marginal distribution functions of variables $X$ and $Y$, respectively, and $F_{X,Y}(x,y)$ is the joint distribution function of the two-dimensional variables $(X,Y)$.

### 3.3.3 Combination scenario risk probability

To carry out the tropical cyclone storm surge and wave combination scenario simulation, we classify the SH and SWH into five classes (Table 2) by referring to the *Technical directives for risk assessment and zoning of marine disasters—Part 1: Storm Surge* (MNR, 2019) and *Part 2: Waves* (MNR, 2021). Based on the marginal and copula functions of the storm surge and wave, we calculate the bivariate risk probabilities for different hazard level combination scenarios. The calculation formula is expressed as follows:





$$P(x_1 < X \le x_2, y_1 < Y \le y_2) = P(X \le x_2, Y \le y_2) - P(X \le x_2, Y \le y_1) - P(X \le x_1, Y \le y_2) + P(X \le x_1, Y \le y_1) = F_{X,Y}(x_2, y_2) - F_{X,Y}(x_2, y_1) - F_{X,Y}(x_1, y_2) + F_{X,Y}(x_1, y_1) \tag{9}$$

**Table 2 Hazard level classification criteria for combined scenarios of TC surge height and significant wave height**

| Level | Surge height (m) | Significant wave height (m) |
|---|---|---|
| I | [2.5, +∞) | [14.0, +∞) |
| II | [2.0, 2.5) | [9.0, 14.0) |
| III | [1.5, 2.0) | [6.0, 9.0) |
| IV | [1.0, 1.5) | [4.0, 6.0) |
| V | [0.0, 1.0) | [0.0, 4.0) |

### 3.4 Design of protection standards for storm surge and wave

### 3.4.1 Risk probability changes under increased storm surge and wave protection standards

In actual engineering protection design, if the protection standards of SH and SWH are appropriately increased or decreased,

it can change the simultaneous bivariate risk probability $P_\cap$, joint bivariate risk probability $P_\cup$, and conditional bivariate risk

probability $P_|$. In this paper, we try to estimate the change value of the bivariate risk probability by raising the return period

protection standard of storm surge or wave scenarios. The formula is expressed as follows:

$$P_{d_\cap} = P\big((X > x_2) \cap (Y > y)\big) - P\big((X > x_1) \cap (Y > y)\big)$$
$$= P(X \le x_2, Y \le y) - P(X \le x_2) - P(X \le x_1, Y \le y) + P(X \le x_1) \tag{10}$$
$$= F_{X,Y}(x_2, y) - F_X(x_2) - F_{X,Y}(x_1, y) + F_X(x_1)$$

$$P_{d_\cup} = P\big((X > x_2) \cup (Y > y)\big) - P\big((X > x_1) \cup (Y > y)\big) = P(X \le x_1, Y \le y) - P(X \le x_2, Y \le y)$$
$$= F_{X,Y}(x_1, y) - F_{X,Y}(x_2, y) \tag{11}$$

$$P_{d_|} = P\big((X > x_2)|(Y > y)\big) - P\big((X > x_1)|(Y > y)\big)$$
$$= \frac{P(X \le x_2, Y \le y) - P(X \le x_2) - P(X \le x_1, Y \le y) + P(X \le x_1)}{1 - P(Y \le y)} \tag{12}$$
$$= \frac{F_{X,Y}(x_2, y) - F_X(x_2) - F_{X,Y}(x_1, y) + F_X(x_1)}{1 - F_Y(y)}$$

where $P_{d_\cap}$, $P_{d_\cup}$, and $P_{d_|}$ are the reduced values of the simultaneous risk probability $P_\cap$, the joint risk probability $P_\cup$, and

the conditional risk probability $P_|$ after the univariate return period protection standard is raised, and $x_1$ and $x_2$ are the



intensity values of variable $X$ for different return periods, respectively, where $x_2 > x_1$.

### 3.4.2 Design storm surge and wave criteria for joint return period scenarios

The marginal and joint probabilities of storm surge and wave scenarios cannot be directly employed as reference values for engineering protection standards. Therefore, we explore the design criteria for the combined storm surge and wave scenarios based on the joint return periods $RP_\cup$ and simultaneous return periods $RP_\cap$. Under the constraint that the joint return period

of surge height and significant wave height is $K$, the maximum simultaneous bivariate risk probability is selected as the objective function. This is the case where the bivariate risk probability is the maximum considering the correlation of two disaster-causing factors, and the corresponding simultaneous return period is the smallest, which is the most appropriate case for prevention. Therefore, the optimal design criteria for storm surge and wave scenarios are estimated using the nonlinear programming method. The formula is expressed as follows:

Constraint condition:

$$\begin{cases} K = RP_\cup = \dfrac{E_L}{P\big((X > x) \cup (Y > y)\big)} = \dfrac{E_L}{1 - P(X \le x, Y \le y)} = \dfrac{E_L}{1 - F_{X,Y}(x, y)} \\ \qquad\qquad x \in (0, 40) \\ \qquad\qquad y \in (0, 40) \end{cases} \tag{13}$$

Objective function:

$$max\{P_\cap\} = min\{RP_\cap\} = min\left\{\dfrac{E_L}{P\big((X > x) \cap (Y > y)\big)}\right\} = min\left\{\dfrac{E_L}{1 - F_X(x) - F_Y(y) + F_{X,Y}(x, y)}\right\} \tag{14}$$

## 4 Results and discussion

### 4.1 Optimal marginal function

Since the density and location of the triangular network in the storm surge model and wave model differ, we use the nodes of the storm surge triangular network as the benchmark and use the wave node closest to each storm surge node as the wave simulation result based on the nearest neighbor method. Therefore, a dataset of storm surges and waves with the same number

and location of nodes is reconstructed, containing 1665 nodes in the study area.

In this paper, based on the reconstructed storm surge and wave simulation results of historical TC events, we calculate the





annual extremes of SH and SWH for each node. Next, the time series of the bivariate annual maximum value for all nodes are fitted with marginal functions by five functions, including Gumbel, Weibull, gamma, exponential, and generalized extreme value distribution (GEV), and the fitting effect is judged by the K-S test to determine whether it passes the 95% significance level. Next, we count the frequency of each type of function passing the K-S test in all nodes and its percentage of the total number of nodes (Table 3).

**Table 3 Frequency and percentage of five functions passing the K-S test for all nodes of SH and SWH**

| Marginal function | Surge height | | Significant wave height | |
|---|---|---|---|---|
| | Frequency | Percentage (%) | Frequency | Percentage (%) |
| Gamma | 1508 | 90.57 | 1464 | 87.93 |
| Exponential | 1567 | 94.11 | 1076 | 64.62 |
| Gumbel (right) | 1615 | 97.00 | 1629 | 97.84 |
| Weibull (max) | 1469 | 88.23 | 300 | 18.02 |
| GEV | 1665 | 100.00 | 1657 | 99.52 |



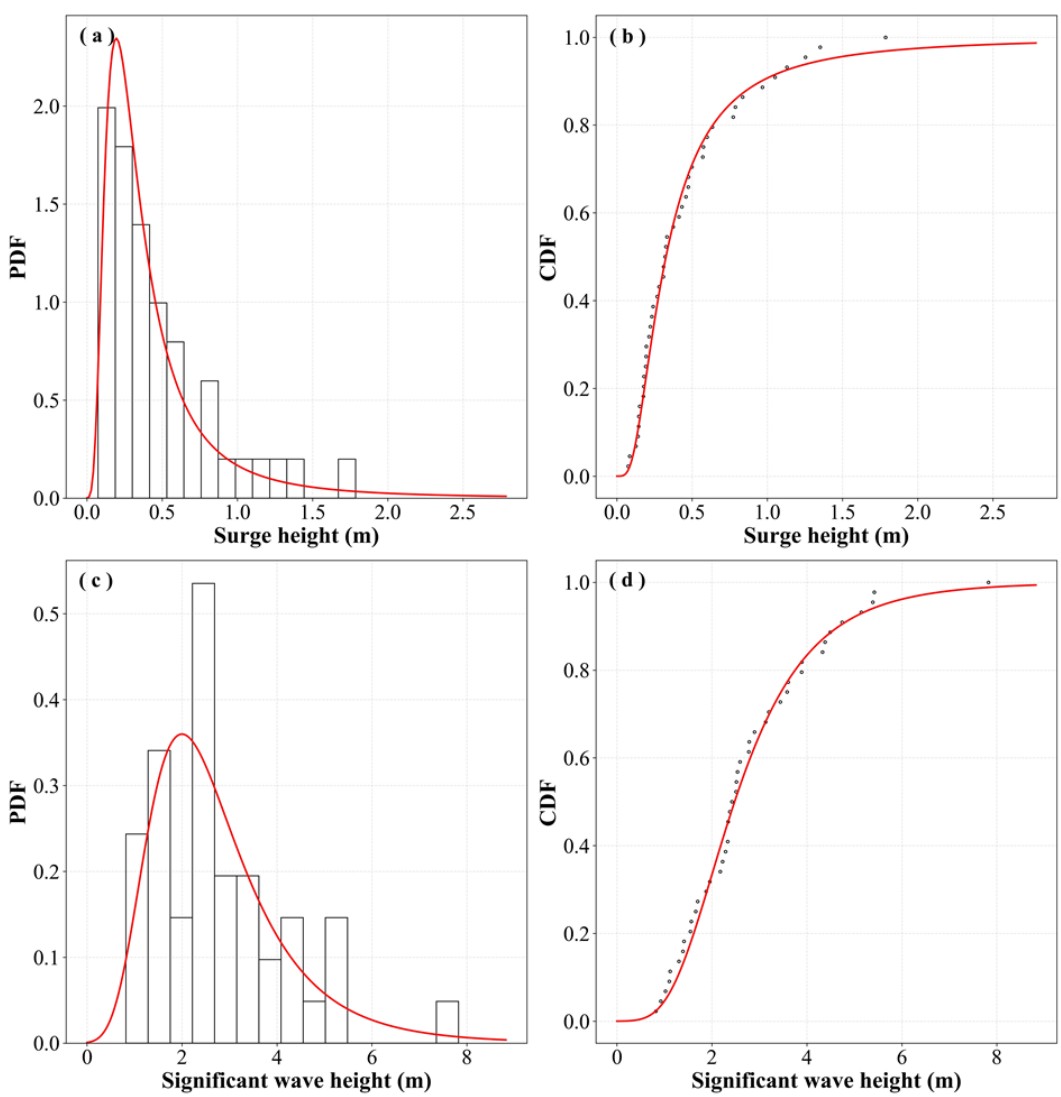

**Figure 3: Fitting results of the PDF and CDF of the SH and SWH based on the GEV function (using node (110.5142° E, 20.2768° N)**

**as an example)**

Based on the statistical results, it is determined that the GEV function has the highest K-S test pass rate of 100% for the

univariate fitting of SH, so GEV is set as the optimal marginal function in this study. For SWH fitting, the number of nodes

passing the K-S test for the GEV function is 1657, accounting for 99.52% of the total number of nodes, which is higher than

other fitting functions. We apply the GEV function to fit the marginal function of the SH and SWH at all nodes and calculate

the PDF, CDF, and RP. Figure 3 shows an example of the PDF and CDF of the SH and SWH for a node.

## 4.2 Distribution of univariate return periods

Based on the univariate return period formula (Eq. 2), the SH and SWH are estimated for six typical return periods of 5a, 10a, 20a, 50a, 100a, and 200a at all nodes. To analyze the study area's univariate return period distribution characteristics, we chose the cubic spline interpolation method to interpolate the intensity values at each node with different return periods into a raster

with a resolution of 1 km (Figure 4 and Figure 5).

As shown in Figure 4, the SH shows a significant increasing trend as it approaches the coastline. In general, the SH along the eastern coast of the Leizhou Peninsula is higher than that in other regions, mainly influenced by factors such as TC landing location, landing direction, and pocket-shaped coastal topography. The Northern Hemisphere TCs are counterclockwise rotations affected by the deflecting force. Therefore, northeastern Hainan Island is located on the southern coast of the

Qiongzhou Strait, and the northeast and northwest winds of TCs affecting the region easily cause seawater accumulation, which is a storm surge-prone area. The TC wind fields in the east and south of Hainan Island are not as favorable to water gain as those in the north. Therefore, the offshore surge height on Hainan Island shows a high distribution in the northeast and a low distribution in the southeast. With an increase in the return period, the SH in each region offers different degrees of growth, and the regional divergence is more prominent, among which there is an obvious area of increase in southeastern Hainan Island

because the region is located at the edge of the continental shelf and the seafloor topography is highly variable.



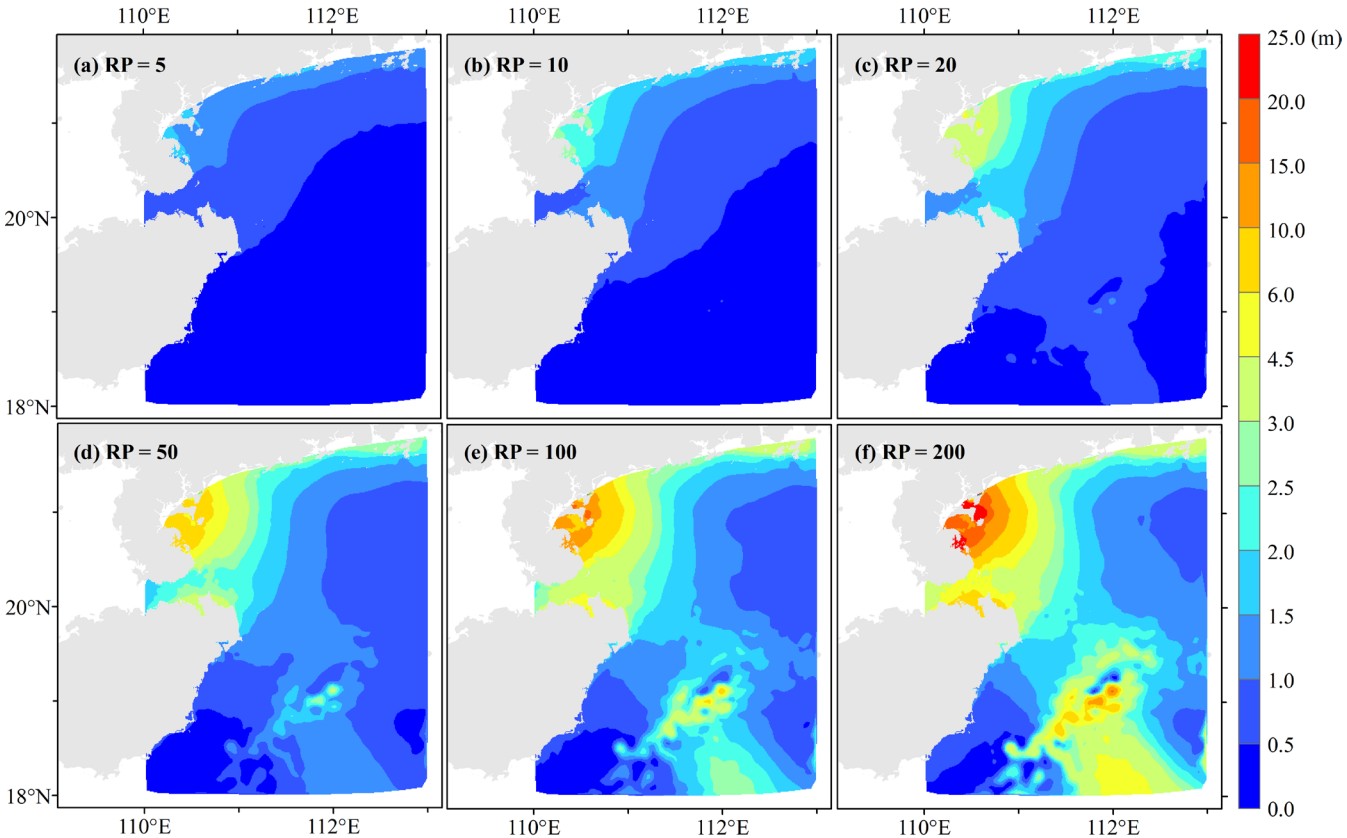

**Figure 4: Spatial distribution of surge heights of tropical cyclones for six typical return periods**

As shown in Figure 5, the SWH near the shore is generally smaller than that in the deep sea and offers a significant decreasing trend as it approaches the coastline. This finding is mainly attributed to the shallow shore depth, island obstruction, wave breaking, and seabed friction attenuation. Among them, the SWH in the eastern Leizhou Peninsula is lower than that of other seas and is influenced by the curved and concave coastline and the topography of the shore section. The SWH in the east and south of Hainan Island is high because of the wave-breaking effect and dissipation caused by the dramatic change in water depth in the region, so there is a more significant gradient. The SWH north of Hainan Island is low, and the shift from sea to land is relatively slow. In general, the SWH increases with an increasing return period, and the regional differences are better in each region.




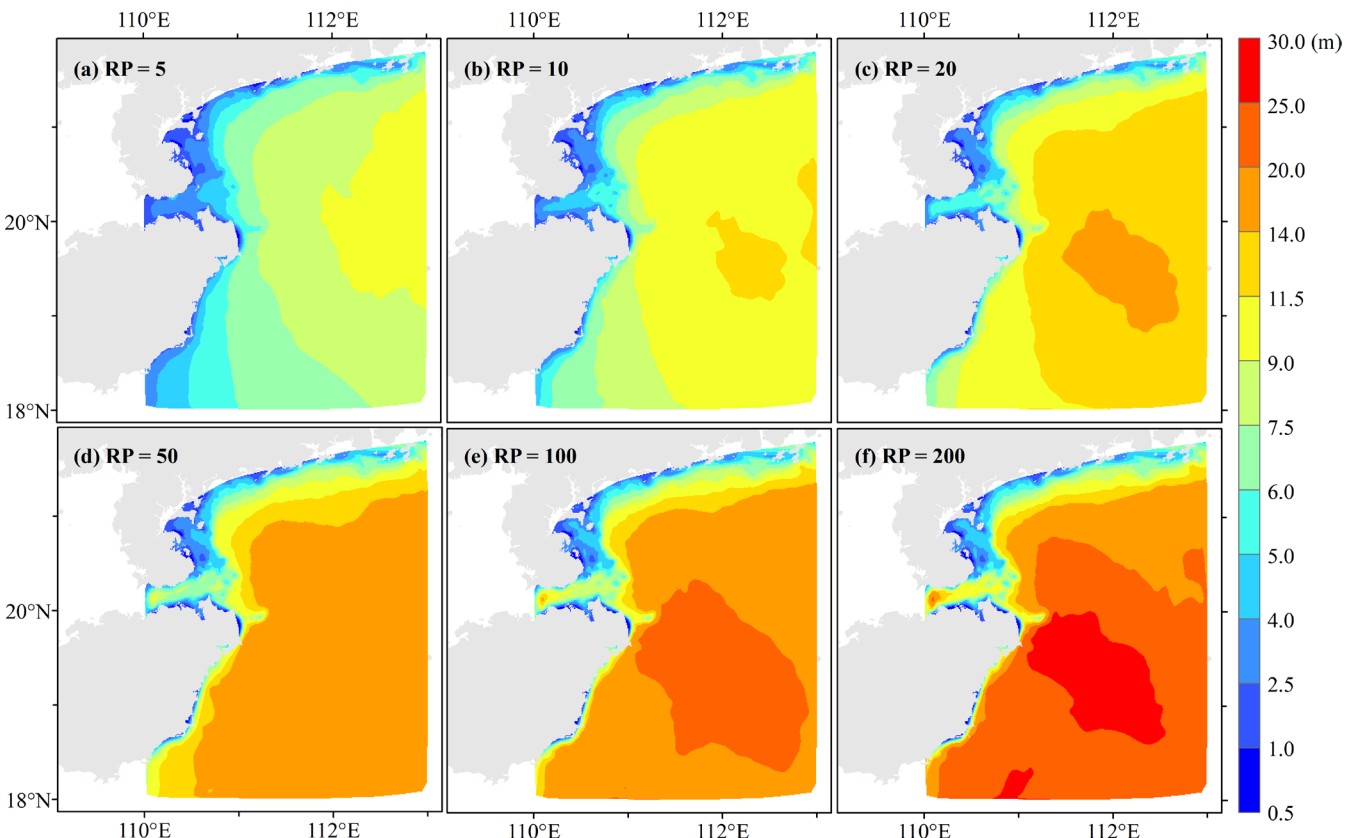

**Figure 5: Spatial distribution of significant wave heights of tropical cyclones for six typical return periods**

## 4.3 Optimal copula function

The optimal GEV function is utilized as the marginal function for the TC storm surges and waves, based on which three copula

functions are applied to the bivariate joint fitting of 1665 nodes. The function parameters are fitted by the maximum likelihood

method, and the K-S test is used to determine whether the fit passes the 95% significance level. Next, we count the number of

nodes that pass the K-S test for the three types of copula functions and their percentage of the total number of nodes (Table 4).

The statistical results show that the number of nodes passing the K-S test for the Gumbel copula function is 1603, accounting

for 96.28% of all nodes, so it is used as the optimal copula function in this study. The Gumbel copula function is applied to the

bivariate joint fitting of SH and SWH for all nodes, and the PDF and CDF are calculated. Figure 6 shows an example of the

PDF and CDF of a node's optimal copula function for SH and SWH.



**Table 4 Frequency and percentage of three two-dimensional copula functions passing the K-S test for all nodes of surge height and significant wave height of tropical cyclones**

| Copula function | Frequency | Percentage (%) |
| --- | --- | --- |
| Clayton | 486 | 29.19 |
| Frank | 1398 | 83.96 |
| Gumbel | 1603 | 96.28 |

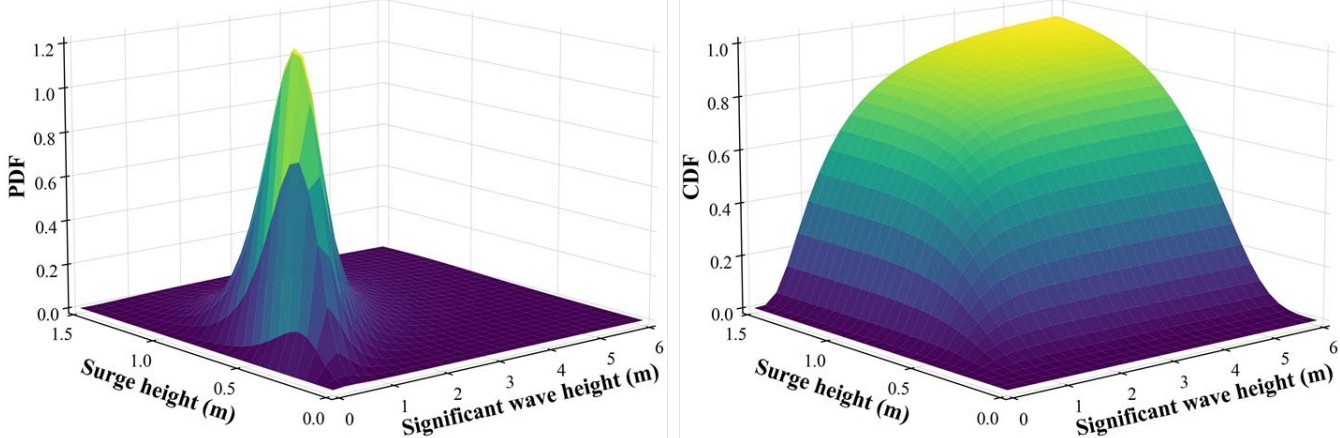

**Figure 6: Fitting results of PDF and CDF for two-dimensional joint tropical cyclone surge height and significant wave height based on the Gumbel copula function (using the node (110.5142° E, 20.2768° N) as an example)**

**4.4 Distribution of bivariate risk probabilities and return periods**

Based on the optimal marginal distribution function and copula function, we calculate $RP_\cap$, $RP_\cup$, and $RP_|$ of SHs and SWHs. When the intensity values of the two disaster-causing factors are equivalent, $RP_\cap$ is greater than $RP_\cup$, which indicates that $P_\cap$ is smaller than.$P_\cup$. Figure 7 shows an example of $RP_\cap$ and $RP_\cup$ for the SH and SWH at a node based on the optimal copula function.





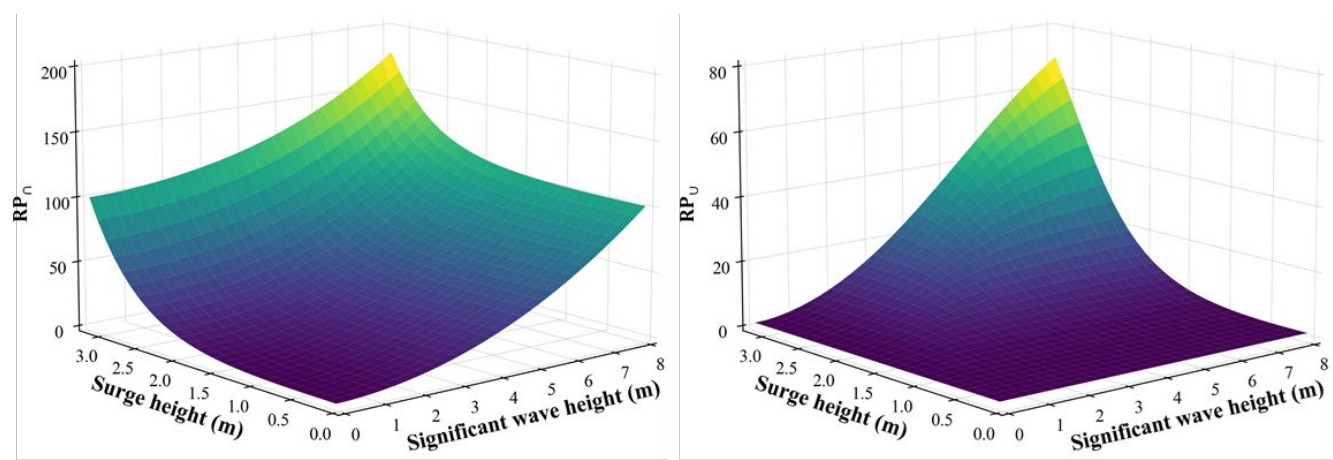

**Figure 7: Distribution of the simultaneous return period ($RP_\cap$) and joint return period ($RP_\cup$) of the tropical cyclone surge height and significant wave height based on the Gumbel copula function**

Based on the formula of bivariate risk probability (Eq. 3 and Eq. 5), $P_\cap$ and $P_\cup$ of SH and SWH are calculated for all nodes

with four typical combinations of return periods of 10a, 20a, 50a, and 100a. To analyze the distribution characteristics, $P_\cap$

and $P_\cup$ for different combinations of return periods at each node are interpolated into a raster with a resolution of 1 km using

the cubic spline interpolation method (Figure 8 and Figure 9).

Bivariate $P_\cap$ gradually decreases as the return period of SH or SWH increases (Figure 8). In general, the closer to the coastline,

the higher $P_\cap$. $P_\cap$ is greatest when the return period of SH and SWH is 10a, which is higher than 0.05. $P_\cap$ is the smallest

for SH and SWH of 100a, which is generally lower than 0.009.





**Figure 8: Simultaneous risk probabilities of combined scenarios with four typical return periods for surge height and significant wave heights of tropical cyclones**

$P_{\cup}$ of SH and SWH is higher than $P_{\cap}$, and it gradually decreases with an increasing return period of the two disaster-causing

factors (Figure 9). Overall, the closer to the coastline, the higher $P_{\cup}$. $P_{\cup}$ is highest when the return period of SH and SWH

are 10a, which is greater than 0.13 overall. $P_{\cup}$ is smallest when the return period of SH and SWH are 100a, which is less than

0.015. When the return period of SH or SWH is 50a or 100a, the regional differences in $P_{\cup}$ are relatively small.



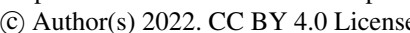

**Figure 9: Joint risk probabilities of combined scenarios with four typical return periods for surge height and significant wave heights of tropical cyclones**

Based on the formula of bivariate $P_|$ (Eq. 7), we calculate $P_|$ for all nodes univariate with different return periods for the other variable in four return periods and interpolate them into 1 km raster data using cubic spline interpolation. According to the formula, the calculation results are consistent when the positions of the variables are swapped. Therefore, only $P_|$ for the four return periods of SH in different wave return periods are shown in this paper (Figure 10). When the SWH is a specific





return period, bivariate $P_l$ gradually decreases as the return period of the SH increases. Under the condition that the return period of SWH is 10a, bivariate $P_l$ for SH with a return period of 10a are concentrated between 0.55 and 0.75, and $P_l$ is generally less than 0.08 if the return period for SH is 100a. When the return periods of SWHs and SHs are equivalent, bivariate $P_l$ of are concentrated between 0.55 and 0.75.

**Figure 10: Conditional risk probabilities of bivariate for different return periods of tropical cyclone significant wave heights**


According to the classification criteria of disaster-causing factors (Table 2), SH and SWH are divided into five classes. We calculate two-dimensional $P_U$ based on Eq. 9 for all nodes with different combinations of SH and SWH for a total of 25 scenarios and interpolate them into 1 km raster data using the cubic spline interpolation method (Figure 11).

In terms of the vertical variation pattern, with the determination of the SH level, as the SWH level increases, the high-value area of the combined scenario risk probability gradually moves away from the coastline, and the scope of the nearshore low-value area gradually expands. This result is consistent with the geographic distribution pattern that the SWH is low nearshore and high offshore. In the horizontal variation law, when the SWH level is determined, as the SH level increases, the range of low-value areas for the combined scenario risk probabilities expands, and the low-value area's left boundary gradually

approach the coastline. This result is consistent with the geographic distribution of SHs being high nearshore and low offshore. Overall, the maximum value of the risk probability for each combined scenario tends to decrease as the level of SH or SWH increases. The larger SH and SWH are concentrated in the eastern Leizhou Peninsula at a certain distance offshore, and the remainder of the region is less likely to occur.







**Figure 11: Risk probabilities of combined scenarios with different levels of surge height and significant wave height for tropical cyclones**

Based on the calculated $P_\cap$, $P_\cup$, $P_|$, and $P_\&$ with different return periods, Markov chain Monte Carlo (MCMC) and other methods can be further applied to generate random samples for the quantitative assessment of TC storm surges and waves. On the other hand, we can explore the effect of changes in SH and SWH on the bivariate joint risk and apply it to the design of engineering protection criteria.



### 4.5 Design storm surge and wave criteria

In the design of the engineering fortification criteria, if one disaster-causing factor is dominant, the return period for the other variable is increased under the condition that its return period fortification standard is determined, which can effectively change bivariate $P_\cap$ and $P_\cup$. In this paper, we calculate the change in risk probability based on Eq. 10, Eq. 11, and Eq. 12 to determine

that the shift in risk probability remains the same when the positions of the two disaster-causing factors are switched. Therefore, the change values of $P_\cap$, $P_\cup$, and $P_|$ are calculated for all nodes of SH and SWH for one variable with return periods of 10a, 20a, 50a, and 100a protection criteria when the design return period criteria for the other variable are raised from 5a, 10a, 20a, and 50a to 10a, 20a, 50a, and 100a, respectively. And the data are interpolated into 1 km raster data using the cubic spline interpolation method (Figure 12, Figure 13, and Figure 14).

Figure 12 shows the distribution of the reduced values of bivariate $P_\cap$ for the scenario with elevated univariate return period protection criteria. In general, as the return period protection criteria of one variable increase, the decrease in $P_\cap$ gradually decreases as the return period of the other variable's protection standard increases. Its reduction is concentrated between 0 and 0.035. When the return period protection standard of one variable is fixed, as the protection criteria of another variable are gradually increased, the decline of $P_\cap$ rises to a certain level and then tends to decrease. When the return period of one variable

is 10a or 20a, the decline in $P_\cap$ increases when the protection standard of another variable is raised. If the design criteria increase from 50a to 100a, the change value of $P_\cap$ decreases.



**Figure 12: Difference in the simultaneous risk probability of tropical cyclone surge height and significant wave height for scenarios with elevated return period protection standards**

Figure 13 shows the distribution of the reduced values for bivariate $P_{\cup}$ with the elevated univariate return period protection criteria. In general, $P_{\cup}$ decreases more than $P_{\cap}$, and the reduced value of $P_{\cup}$ varies from 0 to 0.105. As the return period protection standard for one variable gradually increases, $P_{\cup}$ slowly decreases after the protection criteria for the other variable increase. When the return period protection criterion for one variable is fixed, the decline in $P_{\cup}$ gradually decreases as the





design criteria for the other variable are increased.



**Figure 13: Difference in joint risk probability of tropical cyclone surge height and significant wave height for scenarios with elevated return period protection standards**

Figure 14 shows the distribution of the reduced values of bivariate $P_1$ for the scenario of raising the univariate return period protection criteria. As the return period for one variable increases, there is a decreasing trend in the decrease in $P_1$ after the design criteria for the other variable are raised. $P_1$ has a more significant decrease than $P_\cap$ and $P_\cup$, and the decreasing value





of $P_|$ varies from 0 to 0.45. When the protection level of one variable is fixed and low, the reduction in $P_|$ will tend to decrease after the design criteria of another variable are raised to a certain level. When the protection standard for one variable is 10a or 20a, the decrease in bivariate $P_|$ tends to increase when the design criterion for the other variable's return period is raised, but the decrease in $P_|$ is slightly reduced when the design criterion of the other variable is increased from 50a to 100a.

If the protection level of one variable is high, the decrease in $P_|$ after the protection standard of the other variable is raised always tends to increase.





**Figure 14: Differences in the conditional risk probability of tropical cyclone surge height and significant wave height for scenarios with elevated return period protection standards**

In the design of engineering standards, based on $RP_{\cup}$ and $RP_{\cap}$ of SH and SWH to set the appropriate reference value of the protection standards, the estimation method is shown in Section 3.4.2. In this paper, the design values of SH and SWH for six $RP_{\cup}$ for all nodes are calculated based on the above method and interpolated to 1 km raster data by the cubic spline interpolation method (Figure 15 and Figure 16). In general, with an increase in the return period, the design criteria of SH and




SWH show an apparent increasing trend, in which the high-value area of SH is constantly concentrated east of the Leizhou

Peninsula, and the high-value area of SWH is concentrated in the eastern sea area of Hainan Island.

When $RP_U$ is 5a, the design criteria of SH are between 1.5 m and 2.5 m in the eastern coastal area of the Leizhou Peninsula

and fall below 0.5 m in the southeastern coastal region of Hainan Island. As the return period increases, the SH design standard

gradually increases, and when $RP_U$ is 200 a, the SH design standard in the eastern coastal area of the Leizhou Peninsula is

generally higher than 3.0 m. The SH design standard in the northeast coastal area of Hainan Island is mainly between 3.0 m

and 15.0 m, while that in the southeast coastal region of Hainan Island is between 0.5 m and 2.0 m, which is lower than that

in the northeast.

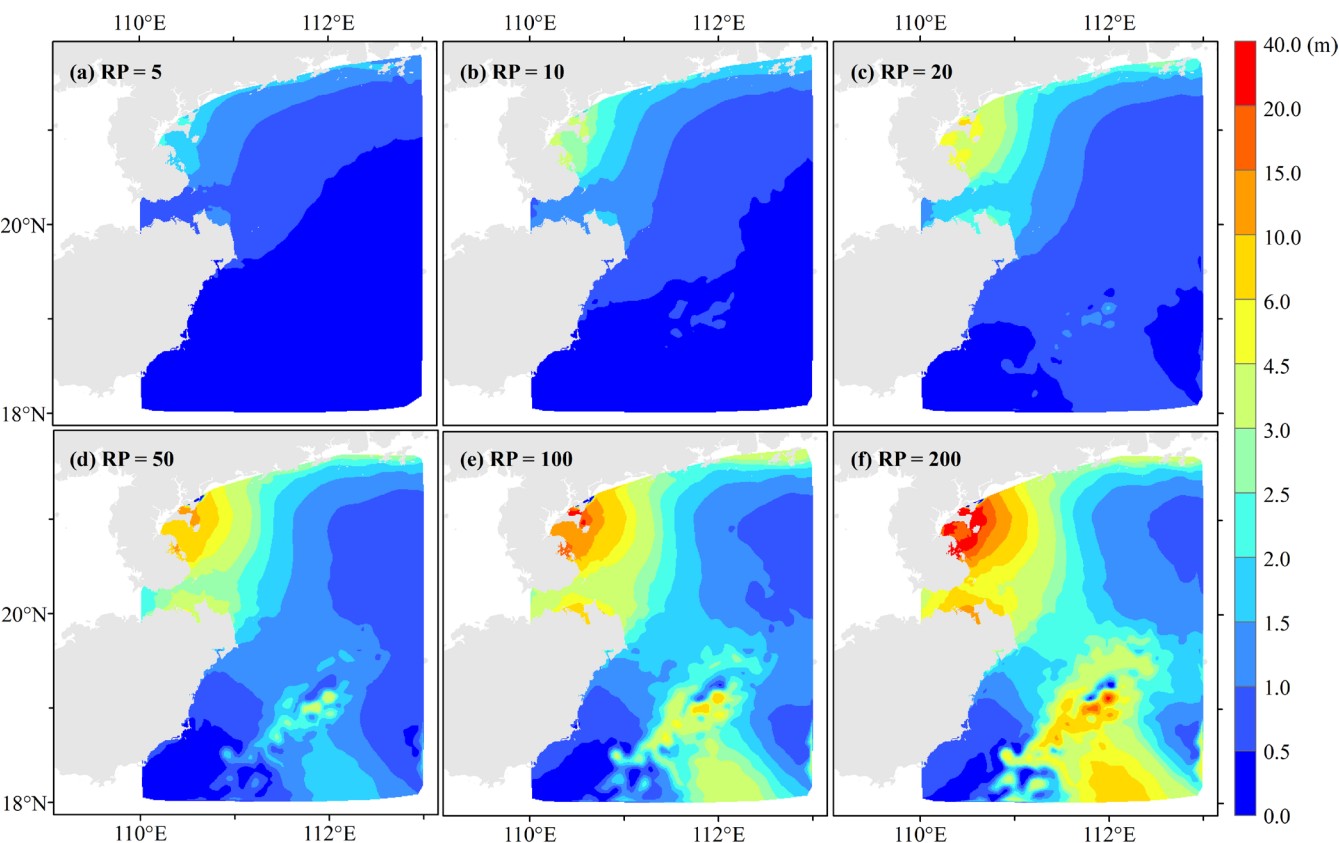

Figure 15: Design values of tropical cyclone surge heights for six typical joint return period scenarios

When $RP_U$ is 5a, the design criteria of SWH in the coastal areas of the Leizhou Peninsula and Hainan Island are mainly less

than 2.5 m. The further from the coastline, the protection standard gradually increases. As the return period increases, the

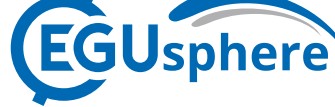

design criteria of SWH gradually increase, and the growth is more evident than that of SH. When $RP_\cup$ is 200 a, SWH along

the coast of the Leizhou Peninsula is generally less than 6.0 m, while the SWH design standard along the Qiongzhou Strait

and southeastern Hainan Island is relatively high.

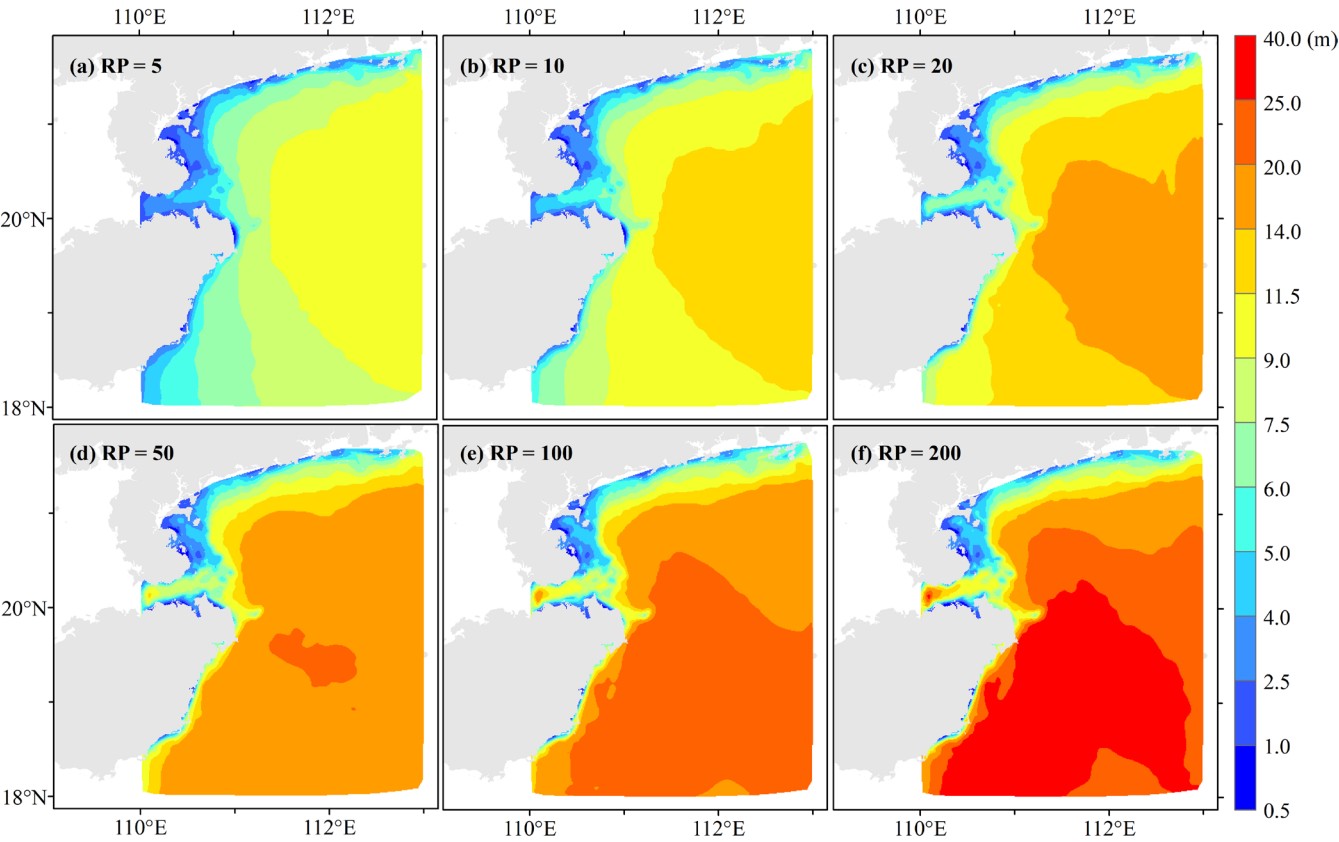

**Figure 16: Design values of tropical cyclone significant wave heights for six typical joint return period scenarios**

**5 Conclusions**

The quantitative assessment of the combined probability distribution characteristics of TC storm surges and waves provides a

basis for compound disaster hazard analysis and the design of engineering fortification criteria, which are essential for disaster

risk management and disaster prevention, mitigation, and relief. In this paper, we explore the joint hazard assessment method

based on the SH and SWH datasets of historical TC events from 1949 to 2013 with a coupled marginal distribution function

and bivariate copula function. The main conclusions are expressed as follows.

1) The GEV function is the best fit for the probability distribution characteristics of the annual extremes of tropical cyclone SH and SWH for all nodes in the study area. The Gumbel copula function is suitable as a bivariate joint distribution function for all nodes in this study area.

2) The hazard of a single disaster-causing factor can be characterized by the univariate intensity values with different return periods estimated by the optimal marginal distribution function. In general, the SH shows a significant increasing trend closer to the coastline, and SWH is higher farther from the shoreline in different return periods. However, the distribution has apparent spatial heterogeneity influenced by the shoreline shape, coastal topography, submarine topography, and deflection forces.

3) Due to the data dimensionality, it is difficult to estimate the exact value of the univariate intensity for a given return period. Therefore, risk probabilities are utilized in this study to assess the joint hazard of multiple disaster-causing factors, including $P_\cap$, $P_\cup$, $P_|$, and $P_\&$, which effectively compensates for the deficiency of disregarding the correlation among variables in univariate hazard assessment. Based on the occurrence probability for different SH and SWH scenarios, many random samples can be generated to provide data for storm surge and wave hazard assessments. These four risk probabilities can visually describe the occurrence probability for different combinations of scenarios; the more significant the risk probability is, the higher the hazard. Overall, $P_|$ is the largest, $P_\cup$ is the second-largest, and $P_\cap$ is the smallest, while $P_\&$ is influenced by the segmentation of the causal factors. When one variable is constant, $P_\cap$, $P_\cup$, and $P_|$ tend to decrease as the return period of the other variable increases.

4) In the design of actual engineering fortification standards, the design criteria of SH or SWH can be appropriately improved to reduce bivariate $P_\cap$, $P_\cup$, and $P_|$. As the return period protection standard of one variable increases, the decline in $P_\cap$ and $P_|$ tends to decrease after the design criterion of another variable is increased, but the decline in $P_|$ gradually increases. When the return period protection standard of one variable is fixed, as the design criteria of another variable gradually increase, the decline in $P_\cap$ and $P_|$ rises to a certain level and then tends to decrease, but the decline in $P_\cup$ gradually decreases. Therefore, the impact of disasters can be effectively reduced by setting appropriate return period design standards for storm surges and waves, which enable coastal areas to cope with a high bivariate risk probability. Since the changing trends of $P_\cap$ and $P_\cup$ are the opposite. These two indicators can be used as constraints and objective functions to estimate the most suitable storm surge and wave protection standards for different $RP_\cup$ values, which provides a scientific reference for optimizing coastal urban

engineering protection standards.

In this paper, the two most commonly employed indicators of SH and SWH are selected for TC storm surge and wave hazard assessment, and the contribution of other indicators to hazard assessment can be explored in subsequent studies. In addition, based on the risk probability distribution estimated in this paper, the random sample generation method can be further explored to simulate SH and SWH in different scenarios, providing more sample data for conducting joint TC storm surge and wave hazard assessments.

*Author contributions.* FWH and ZHX conceived the research framework and developed the methodology. ZHX was responsible for the code compilation, data analysis, graphic visualization, and first draft writing. FWH managed the implementation of research activities and revised the manuscript. CM participated in the data collection of this study. All authors discussed the results and contributed to the final version of the paper.

*Competing interests.* The authors declare that they have no conflict of interest.

*Acknowledgments.* This work was mainly supported by the National Key Research and Development Program of China (grant nos. 2017YFA0604903 and 2018YFC1508803) and the Key Special Project for Introduced Talents Team of Southern Marine Science and Engineering Guangdong Laboratory (Guangzhou) (grant no. GML2019ZD0601). We thank Xing Liu of the Ocean University of China for providing the simulation data of storm surges and waves for historical tropical cyclone events.

*Financial support.* This research has been supported by the National Key Research and Development Program of China (grant nos. 2017YFA0604903 and 2018YFC1508803), and the Key Special Project for Introduced Talents Team of Southern Marine Science and Engineering Guangdong Laboratory (Guangzhou) (grant no. GML2019ZD0601).

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
