# Peer review of "Joint probability analysis of storm surge and wave caused by tropical cyclone for the estimation of protection standard: a case study on the eastern coast of the Leizhou Peninsula and Hainan Island of China"

_EGUsphere, 2022_

## Referee Comment (RC1)

**Joint probability analysis of storm surge and wave caused by tropical cyclone for the estimation of protection standard: a case study on the eastern coast of the Leizhou Peninsula and Hainan Island of China**

**By Z. Haixia, C. Meng, F. Weihua**

**Submitted to NHESS**
*MS-NR: egusphere-2022-847*
* * *
**REFEREE REPORT**

**General comments**

I regret to say that, in my opinion, the quality of this paper is not suitable for publication. Language is poor and technical terms are associated rather randomly resulting in nonsense sentences (e.g. L69, 74, 135, 143, 154, 197, just to mention a few).

Leaving aside the general confusion of the text (denoting lack of familiarity with theoretical concepts), the data analysis is the usual superficial application of bivariate copulas already reported in hundred of papers, iterating widespread mistakes and lacking whatever uncertainty assessment (I cannot understand why uncertainty assessment is fundamental to provide a decent univariate frequency analysis, but it suddenly disappears and is neglected when moving to multivariate frequency analysis, which is affected by the additional uncertainty related to the unknown dependence structure).

Some results that are interpreted as empirical findings are just theoretical constraints that do not provide any insight.

Justifying a paper with sentences like "*Presently, a few scholars have conducted studies on the interaction and joint distribution of storm surges and waves*" makes no sense: the use of joint distributions and copulas have been used in coastal/ocean engineering for at least 15 years worlwide to model not only surge and significant wave height but also other met-ocean variables; these methods have also been incorporated in national guidelines (e.g. UK and Netherlands, to mention a few) for several years, and are subject to ongoing improvement and update. So, please perform a decent preliminary literature review before running out-of-the-shelf computer packages.

**Specific comments**

L17: "*the surge height shows an increasing trend closer to the coastline*" ?? -> the surge height shows higher values closer to the coastline.

L18: "*when one variable is constant, the simultaneous, joint, and conditional risk probability tends to decrease as the other variable increases*"
Marginal and joint distributions are monotonic functions! This behaviour is related to their mathematical properties and has nothing to do with the analysed variables.

L58: "*the study of the joint probability distribution of tropical cyclone storm surges and waves is conducive to improving the accuracy and precision of joint hazard assessment*"
Yes, this is true if we neglect the fact that joint distributions are affected by the same uncertainty of their marginals and the additional uncertainty of the dependence structure.

L69: "*However, the constructed joint distribution is still a probabilistic result, and further search for constraint relations is needed to provide a basis and guidance for disaster prevention and mitigation. Therefore, this paper quantitatively analyzes the occurrence probability of storm surge and wave combinations based on the fitting results of the copula function*"
Words have a meaning and should be used accordingly! First, you say that "constraint relations" are needed because joint distributions are still a probabilistic (which seems to be a minus, whatever that sentence means), and then joint distributions are used neglecting constraints. Those sentences contradict each other.

L74: "*In the design process of sea dikes, breakwaters, and harbors, the surge height and significant wave height in different return periods is separately considered... disregarding the correlation between storm surge and so that the calculated water level may be underestimated or overestimated.*"
This is incorrect, underestimation/overestimation means that an estimate is smaller/larger than a true value. In real-world applications, (i) the true value is unknown, so you never know if an estimator overestimates/underestimates, and (ii) univariate probabilities cannot be compared with joint probabilities as they refer to different processes! The choice of the required probabilistic model depends on the specific problem/failure mechanism. Univariate analysis is perfectly fine if there is a unique target design variable, and it correctly estimates the required probability (these concepts are explained here https://link.springer.com/article/10.1007/s00477-014-0916-1)

L80: "*optimize*" -> fit
As the confusion due to meaningless terminology already affects a significant part of the literature, please, use consistent technical terms without inventing a new vocabulary!

L81: "*function by the passing rate of the K-S test*"
When performing tests, the only meaningful "rate" is the rejection rate! (see here https://www.sciencedirect.com/science/article/pii/S0309170817305845 for an explanation)

L135: If the result of rewording concepts is this one, then it is preferable copying and pasting from some good paper/handbook. These sentences are just a set of randomly chosen words that make no sense whatsoever.

L151: Archimedean, elliptical and quadratic are not families but classes of copulas, and they are not the only existing classes. Archimedean copulas can be multiparametric as well! Please, do not use models without knowing the underlying theory, as the result is just a collection of misconceptions, meaningless statements, and misinterpreted figures and tables.

L154: "*analyze the joint probabilities of the marginal distributions of two variables*" -> "the joint probabilities of two variables" Joint probabilities of the marginal distributions do not exist!

L175: The conditional probabilities and the corresponding conditional distributions and return periods are not joint (multidimensional) but univariate (unidimensional).

L197: "*The marginal and joint probabilities of storm surge and wave scenarios cannot be directly employed as reference values for engineering protection standards*"
Return periods are just indicators derived from probabilities: if the former can be used, the latter can be used as well.

Sect. 3.4.2: this awkward method is quite useless because AND and OR return level curves are complementary, and their surfaces are monotonic and diagonally symmetric for the considered Archimedean copulas. This means that the point estimate (x,y) resulting from the (unnecessary) nonlinear optimization is just the intersection of the main diagonal of the copula and the OR level curve with RP = k. No optimization is required, just some familiarity with the theoretical properties of the models one intends to use.

L214: "determine whether it passes the 95% significance level."
"Passing the 95% significance level" makes not sense. A test can only reject or not reject the reference/null hypothesis at a given significance level. The significance level is not 95% but 5%.

Table 3 and corresponding discussion: Using the rate of no rejection to select the best model makes no sense for several reasons:
1) "No rejection" does not mean acceptance, and if two or more models are not rejected for a given data set, the only possible conclusion is that the information/ data is not enough to discriminate among the models.
2) The comparison includes distributions with different number of parameters (1-parameter Exponential, 2-parameter Gamma, Gumbel and Weibull (assuming that the Weibull does not include a location parameter), and the 3-parameter GEV). So, GEV has the obvious advantage of being more flexible due to higher parameterization.

3) Performing a test on 1665 nodes (i.e., 1665 times) is a multiple testing exercise. Under independence (which is not valid in this case), if the null hypothesis is valid, the number of no rejections has an expected value equal to 1582 (95%), and a binomial distribution, thus meaning that the 95% confidence interval of the number of no rejections is (1564, 1599). Therefore, Exponential and Gumbel cannot be discarded for the variable SH.

   However, met-ocean variables come from a model and refer to grid points of a connected area, they are surely strongly correlated in space. This means that the binomial distribution strongly underestimates the actual variability of the number of no rejections because of information redundancy. Therefore, other distributions for both SH and SWH might not be discarded. This is not surprising as these distributions are fitted to just 60/65 annual maxima, i.e. a very small sample size.

4) "No rejection" rate for Weibull and SWH is reasonably wrong because (i) WH/SWH (in deep water) has been historically modelled by Rayleigh distribution, which a particular case of Weibull, (ii) Weibull was shown to be a very good model in shallow water for WH (see https://doi.org/10.1016/j.coastaleng.2022.104130), and (iii) Weibull and 3-paremeter Weibull are closely related to GEV, and Weibull is also a pre-asymptotic distribution in EVT.

   In summary the selection strategy seems to work just because it neglects the foregoing theoretical aspects.

Figure3: What about complementing these figures with confidence/prediction intervals?

L249: "*In general, the SWH increases with an increasing return period*"
In general? Distributions are monotonic functions: higher quantiles always correspond to higher return period for whatever (continuous) random variable… always, not "in general"!

Fig. 6: This figure (along with Fig. 7) is uninformative. The shape of the surfaces of generic joint PDFs and CDFs is well known. What matters is some diagnostic plot showing the goodness of fit (with uncertainty!)

L270: "*When the intensity values of the two disaster-causing factors are equivalent, $R_\cap$ is greater than $R_\cup$, which indicates that $P_\cap$ is smaller than $P_\cup$.*"
This is not a result but the effect of theoretical constraints (see here https://link.springer.com/article/10.1007/s00477-014-0916-1)

Figs 8-10: These figures just report what is expected according to the monotonic nature of bivariate distributions.

L315: "$P_\&$" was not defined in the text. I guess it refers to the discretised classes in Sect. 3.3.3 and Fig. 11.

L409-410: These statements are not related to the preceding text; they are just generic sentences.

Sincerely,

Francesco Serinaldi

---

## Author Comment (AC1)

**General response:** We sincerely thank the reviewers for their valuable feedback, which we have used to improve the quality of our manuscript. The reviewer comments are provided below in **bold font,** and specific concerns have been numbered. Our responses are given in normal font, and changes/additions to the manuscript are given in blue text.

**Responses to Referee #1**

**General comments**

**I regret to say that, in my opinion, the quality of this paper is not suitable for publication. Language is poor and technical terms are associated rather randomly resulting in nonsense sentences (e.g. L69, 74, 135, 143, 154, 197, just to mention a few).**

**Leaving aside the general confusion of the text (denoting lack of familiarity with theoretical concepts), the data analysis is the usual superficial application of bivariate copulas already reported in hundred of papers, iterating widespread mistakes and lacking whatever uncertainty assessment (I cannot understand why uncertainty assessment is fundamental to provide a decent univariate frequency analysis, but it suddenly disappears and is neglected when moving to multivariate frequency analysis, which is affected by the additional uncertainty related to the unknown dependence structure).**

**Some results that are interpreted as empirical findings are just theoretical constraints that do not provide any insight.**

**Justifying a paper with sentences like "*Presently, a few scholars have conducted studies on the interaction and joint distribution of storm surges and waves*" makes no sense: the use of joint distributions and copulas have been used in coastal/ocean engineering for at least 15 years worldwide to model not only surge and significant wave height but also other met-ocean variables; these methods have also been incorporated in national guidelines (e.g. UK and Netherlands, to mention a few) for several years, and are subject to ongoing improvement and update. So, please perform a decent preliminary literature review before running out-of-the-shelf computer packages.**

> **Response:** Your comments and suggestions are appreciated. We believe most of them are valuable to the improvement of this manuscript, though we are unable to agree with all of them. Kindly take your time and find our detailed response and revisions below.

**Specific comments**

1. **L17: "*the surge height shows an increasing trend closer to the coastline*" ?? -> the surge height shows higher values closer to the coastline.**

   **Response:** Thanks for the suggestion on language improvement.

   **Line 17-18:** Second, the surge height shows higher values as locations get closer to the coastline, and the significant wave height becomes greater further from the coastline.

2. **L18: "*when one variable is constant, the simultaneous, joint, and conditional risk probability tends to decrease as the other variable increases.*" Marginal and joint distributions are monotonic functions! This behaviour is related to their mathematical properties and has nothing to do with the analysed variables.**

   **Response:** Sure it is. But even when we have a function as simple as $y = ax + b$, and suppose a is positive, we may still want to emphasize that y will increase as x increases. So, in many cases, it makes perfect sense, isn't it? We did make slight revisions to avoid misunderstanding, as follows.

   **Line 18-21:** Third, the marginal and bivariate cumulative distribution is monotone increasing functions. Correspondingly, the simultaneous, joint, and conditional probabilities decrease monotonically. Therefore, improving the protection standard for either variable can effectively reduce the bivariate probability in the engineering design.

3. **L58: "*the study of the joint probability distribution of tropical cyclone storm surges and waves is conducive to improving the accuracy and precision of joint hazard assessment.*" Yes, this is true if we neglect the fact that joint distributions are affected by the same uncertainty of their marginals and the additional uncertainty of the dependence structure.**

   **Response:** Uncertainty is of great importance when we quantify their joint probability. But even with the existence of uncertainty or even greater uncertainty, which no one denies, joint probability analysis can still provide valuable insights. The text was revised as below, with uncertainty mentioned.

   **Line 46-49:** The interaction, concurrency, or chain of hazards makes the comprehensive hazard assessment of compound hazards with significant uncertainties. In addition, in the design process of sea dikes, breakwaters, and harbors, the surge height and significant wave height in different return periods are often separately considered (MWR, 2014; MOT, 2015, 2018), disregarding the correlation between storm surge and waves.

   **Line 50-51:** Therefore, the joint probability distribution of tropical cyclone storm surges and waves and their uncertainty analysis are crucial in the integrated assessment of joint hazard intensity (Xu et al., 2022).

4. **L69: "*However, the constructed joint distribution is still a probabilistic result, and further search for constraint relations is needed to provide a basis and guidance for disaster prevention and mitigation. Therefore, this paper quantitatively analyzes the***

*occurrence probability of storm surge and wave combinations based on the fitting results of the copula function*." **Words have a meaning and should be used accordingly! First, you say that "constraint relations" are needed because joint distributions are still a probabilistic (which seems to be a minus, whatever that sentence means), and then joint distributions are used neglecting constraints. Those sentences contradict each other.**

**Response:** We reorganized the text as follows, expressing that, even with a joint probability distribution (a 3d surface or a 2d curve with the probability of one variate fixed) available, it is not enough for setting design criteria. And what needs to do furtherly is to find a specific combination of surge and wave probability, and obtain their respective SH and SWH (i.e., two scalar values).

Please also refer to comment # 22, which was supposed to explain the same thing.

Line 59-60: However, as the bivariate joint probability is a three-dimensional surface, the intercepted curve with the specified probability or one variable fixed is a curve rather than a specific scalar, which is not enough for setting the protection criteria.

5. **L74: "*In the design process of sea dikes, breakwaters, and harbors, the surge height and significant wave height in different return periods is separately considered… disregarding the correlation between storm surge and so that the calculated water level may be underestimated or overestimated.*" This is incorrect, underestimation/overestimation means that an estimate is smaller/larger than a true value. In real-world applications, (i) the true value is unknown, so you never know if an estimator overestimates/underestimates, and (ii) univariate probabilities cannot be compared with joint probabilities as they refer to different processes! The choice of the required probabilistic model depends on the specific problem/failure mechanism. Univariate analysis is perfectly fine if there is a unique target design variable, and it correctly estimates the required probability (these concepts are explained here https://link.springer.com/article/10.1007/s00477-014-0916-1).**

**Response:** (1) For sure, it is not easy or even practically impossible to estimate the so-called TRUE value of the intensity for a given probability. But it is also fair and safe to say that, without considering their interaction, it is highly possible that the TRUE value will be underestimated or overestimated, since the physical interaction mechanism between surge and wave is there. (2) On the other hand, the value of univariate analysis and its application of course are there, which is not the focus of this manuscript. That said, we only want to emphasize the importance of joint probability in the text. But we did revise the sentence a little bit to avoid misunderstanding.

Line 44-49: However, tropical cyclone disasters in coastal areas are caused by the combined effect of storm surges, waves, and other hazards, and the mechanism of disaster generation is very complex. The interaction, concurrency, or chain of hazards makes the comprehensive hazard assessment of compound hazards with significant uncertainties. In addition, in the design process of sea dikes, breakwaters, and harbors, the surge height and significant wave height in different return periods are often separately considered (MWR,

2014; MOT, 2015, 2018), disregarding the correlation between storm surge and waves.

6. **L80: "*optimize*" -> fit. As the confusion due to meaningless terminology already affects a significant part of the literature, please, use consistent technical terms without inventing a new vocabulary!**

**Response:** Relax, but we did not mean to create a new term, and we did not create a new term. When we use the word "optimize," we mean to select the optimal marginal distribution function and proper copula function through the AIC, BIC, and goodness of fit of the K-S test for all nodes, not just a simple fitting process. In order to clarify that, the text was revised as follows.

**Line 61-66:** The goal of this paper is to explore the joint probability characteristics of tropical cyclone storm surges and waves, and to apply the joint probability distribution surface to investigate the methods and steps for the design values of the protection criteria for storm surges and waves under combined scenarios. First, based on historical tropical cyclone surge heights and significant wave height, we fit the marginal distribution and copula function of nodes in the study area, and we use the maximum likelihood method to estimate the parameters. Then, the optimal functions are selected based on the Kolmogorov-Smirnov (K-S) test, AIC, and BIC for all nodes.

7. **L81: "*function by the passing rate of the K-S test.*" When performing tests, the only meaningful "rate" is the rejection rate! (see here https://www.sciencedirect.com/science/article/pii/S0309170817305845 for an explanation)**

**Response:** Thanks for your comment and suggestion. The term "passing rate" was misused in the original text, and it did not refer to the results of the K-S test, and we have changed the original statement in the revised manuscript. In the process of selecting the optimal marginal distribution function and copula function, the goodness-of-fit of each marginal distribution function and copula function is calculated by the K-S test for each node. If the p-value of the K-S test of a node is greater than 0.05, the hypothesis of "the sample obeys a certain theoretical distribution" is not rejected, which means that the node passes the K-S test. Then, the ratio of the number of nodes passing the K-S test to the number of all nodes is calculated, and based on this, the optimal function is determined by combining the AIC, BIC, and D-values of the K-S tests.

**Line 63-66:** First, based on historical tropical cyclone surge heights and significant wave height, we fit the marginal distribution and copula function of nodes in the study area, and we use the maximum likelihood method to estimate the parameters. Then, the optimal functions are selected based on the Kolmogorov-Smirnov (K-S) test, AIC, and BIC for all nodes.

8. **L135: If the result of rewording concepts is this one, then it is preferable copying and pasting from some good paper/handbook. These sentences are just a set of randomly chosen words that make no sense whatsoever.**

**Response:** Thanks for the suggestion on language.

**Line 122-127:** Sklar's theorem (Sklar, 1973) elucidates the role that copulas play in the relationship between multivariate distribution functions and their univariate margins and states that any multivariate joint distribution can be described by a univariate marginal distribution function and a couple describing the dependence structure between the variables (Nelsen, 2006). Let $F(x)$ and $G(y)$ be the marginal distributions of $x$ and $y$, $C$ is the copula, and $H(x,y) = C(F(x), G(y))$, where $H$ is the bivariate joint distribution function of $x$ and $y$ (Serinaldi, 2015). Therefore, the copula function is widely utilized in multi-hazard joint probability analysis of natural disasters (Chen et al., 2019; Lee et al., 2013).

9. **L151: Archimedean, elliptical and quadratic are not families but classes of copulas, and they are not the only existing classes. Archimedean copulas can be multiparametric as well! Please, do not use models without knowing the underlying theory, as the result is just a collection of misconceptions, meaningless statements, and misinterpreted figures and tables.**

**Response:** We believe this is a language problem, which has been revised by rewording. Joint probability analysis is not something like unsolved mathematical puzzles. And we believe avoiding subjective judgment with no consolidated evidence will improve the quality of the review.

**Line 143-146:** Many families of copulas exist and mainly include the following: Meta-elliptical copulas (normal and t), Archimedean copulas (Clayton, Gumbel, Frank, and Ali-Mikhail-Haq), Extreme Value copulas (Gumbel, Husler-Reiss, Galambos, Tawn, and t-EV), and other families (Plackett and Farlie-Gumbel-Morgenstern). Among the various families of copulas, the Archimedean copula is more popular for hydrologic applications (Chen and Guo, 2019).

10. **L154: "*analyze the joint probabilities of the marginal distributions of two variables*" -> "the joint probabilities of two variables". Joint probabilities of the marginal distributions do not exist!**

**Response:** Thanks for the comments. Revision is made as follows.

**Line 146-149:** The commonly employed Archimedean copula functions include Gumbel, Clayton, and Frank (Table 1), which are selected to analyze the joint probabilities of two variables, the surge height and significant wave height of tropical cyclone, and then the maximum likelihood method is used to estimate the parameters of the copula function.

11. **L175: The conditional probabilities and the corresponding conditional distributions and return periods are not joint (multidimensional) but univariate (unidimensional).**

**Response:** Thanks for your comment. But we do not think your comment is correct, especially in the context of this manuscript.

We believe that the conditional probabilities and return periods can be bivariate distributions.

The concept of the conditional probability of a multidimensional random variable is the

probability of $(X > x)$ occurring under the condition that $(Y > y)$, which is calculated by the formula $P_| = P\big((X > x)\big|(Y > y)\big) = \frac{P(X>x,Y>y)}{P(Y>y)}$. Therefore, it is a conditional distribution of the two-dimensional variable $(X, Y)$. In addition, a preliminary introduction to the concepts of multivariate probabilities and return periods is provided in the paper (https://link.springer.com/article/10.1007/s00477-014-0916-1), which also states that conditional probabilities and conditional return periods are multivariate.

12. **L197: "*The marginal and joint probabilities of storm surge and wave scenarios cannot be directly employed as reference values for engineering protection standards.*" Return periods are just indicators derived from probabilities: if the former can be used, the latter can be used as well.**

    **Response:** According to your suggestion, we have deleted "marginal" from the original manuscript and only proposed that the joint probability cannot be used as a reference value.

    This is mainly because the intensity values of storm surges and waves under different univariate return periods can be used as a reference for marine engineering protection. However, the bivariate joint probability/return period is a three-dimensional surface. The intercepted curve under the specified occurrence probability or return period is a curve, so it is difficult to be used directly as a reference value for engineering protection criteria.

    **Line 191-193:** The joint probabilities of storm surge and wave scenarios is hard to be directly employed as reference values for engineering protection standards, since the bivariate joint probability/return period is a three-dimensional surface, and the intercepted curve under the specified occurrence probability or return period is a curve, not a scalar.

13. **Sect. 3.4.2: this awkward method is quite useless because AND and OR return level curves are complementary, and their surfaces are monotonic and diagonally symmetric for the considered Archimedean copulas. This means that the point estimate (x, y) resulting from the (unnecessary) non-linear optimization is just the intersection of the main diagonal of the copula and the OR level curve with RP = k. No optimization is required, just some familiarity with the theoretical properties of the models one intends to use.**

    **Response:** Thanks for your comments. But we have a different opinion from you, and no revision has been made.

    We believe it is essential to use a non-linear optimization method to estimate optimal design values for storm surge and wave combinations. This is because although $RP_\cap$ and $RP_\cup$ are complementary and the three-dimensional surface is monotonic, they are not perfectly diagonally symmetric. Therefore it is not the intersection of the main diagonal and $RP_\cup = K$ that corresponds to the largest value of $RP_\cap$.

14. **L214: "*determine whether it passes the 95% significance level.*" "Passing the 95% significance level" makes not sense. A test can only reject or not reject the reference/null hypothesis at a given significance level. The significance level is not 95% but 5%.**

**Response:** Thanks for your comments. We have now changed the wording to "*the p-value of K-S test is used to determine whether the hypothesis that the sample obeys a certain theoretical distribution is rejected*" in the revised manuscript.

**Line 208-211:** Next, the time series of the bivariate annual maximum value for all nodes are fitted with marginal functions by five functions, including Gumbel, Weibull, gamma, exponential, and generalized extreme value distribution (GEV), and the p-value of K-S test is used to determine whether the hypothesis that the sample obeys a certain theoretical distribution is rejected.

15. **Table 3 and corresponding discussion: Using the rate of no rejection to select the best model makes no sense for several reasons:**

    **1) "No rejection" does not mean acceptance, and if two or more models are not rejected for a given data set, the only possible conclusion is that the information/data is not enough to discriminate among the models.**

    **2) The comparison includes distributions with different number of parameters (1-parameter Exponential, 2-parameter Gamma, Gumbel and Weibull (assuming that the Weibull does not include a location parameter), and the 3-parameter GEV). So, GEV has the obvious advantage of being more flexible due to higher parameterization.**

    **3) Performing a test on 1665 nodes (i.e., 1665 times) is a multiple testing exercise. Under independence (which is not valid in this case), if the null hypothesis is valid, the number of no rejections has an expected value equal to 1582 (95%), and a binomial distribution, thus meaning that the 95% confidence interval of the number of no rejections is (1564, 1599). Therefore, Exponential and Gumbel cannot be discarded for the variable SH.**

    **However, met-ocean variables come from a model and refer to grid points of a connected area, they are surely strongly correlated in space. This means that the binomial distribution strongly underestimates the actual variability of the number of no rejections because of information redundancy. Therefore, other distributions for both SH and SWH might not be discarded. This is not surprising as these distributions are fitted to just 60/65 annual maxima, i.e. a very small sample size.**

    **4) "No rejection" rate for Weibull and SWH is reasonably wrong because (i) WH/SWH (in deep water) has been historically modelled by Rayleigh distribution, which a particular case of Weibull, (ii) Weibull was shown to be a very good model in shallow water for WH (see https://doi.org/10.1016/j.coastaleng.2022.104130), and (iii) Weibull and 3-paremeter Weibull are closely related to GEV, and Weibull is also a pre-asymptotic distribution in EVT.**

    **In summary the selection strategy seems to work just because it neglects the foregoing theoretical aspects.**

    **Response:** Thanks for pointing out the problems in the marginal distribution selection process. We changed/refined the selection process as follows:

(1) The p-value of the K-S test is used to determine whether each node rejects the hypothesis that the sample obeys a certain theoretical distribution (Table 3).

(2) The optimal function is screened by three metrics, AIC, BIC, and the D-value of the K-S test. The smaller the AIC, BIC, and D-value of the K-S test, the better the fit, thus determining the optimal marginal function for each node.

(3) The percentage of each optimal function to the number of all nodes was calculated (Table 3), and the function with the highest rate was selected as the optimal marginal function for the storm surges and waves.

Through the above steps, GEV was finally determined as the marginal function for storm surges and waves:

(1) GEV belongs to 3 parameters, and the parameter fitting is relatively flexible.

(2) The no-rejection numbers of the fitted GEV functions for storm surge and waves are 1665 and 1657, respectively, both of which are in the 95% confidence interval.

(4) Among all functions, the GEV function has the highest percentage of preferences, 30% and 27%, respectively.

**Line 135-140:** In this paper, the maximum likelihood method is used to estimate the fit parameters, based on which the optimal marginal functions for storm surges and waves are screened by the following steps: firstly, the p-value of the K-S test is used to determine whether each node rejects the hypothesis that the samples obey a certain functional distribution; secondly, the optimal function for each node is screened by the three metrics, AIC, BIC and D-value of the K-S test. The smaller the AIC, BIC, and D-value of the K-S test, the better the fit, thus determining the optimal marginal function for each node.

**Line 211-212:** Then, we counted the frequency of each function passing the K-S test and its percentage as well as the frequency of the optimal function and its percentage (Table 3).

**Line 217-221:** Based on the statistical results, it was found that for fitting the SH, the K-S test of the GEV function had the highest no-rejection rate of 100%, and the corresponding preference ratio was 30.04%, so GEV was set as the optimal marginal function in this study. For SWH fitting, the number of nodes with no rejection in the K-S test of the GEV function was 1657, accounting for 99.52% of the total number of nodes, and the corresponding percentage of preferences was also higher than that of other fitting functions.

Table 1 Frequency and percentage of five functions passing the K-S test and the optimal function for all nodes of SH and SWH

| Marginal function | Surge height | | | | Significant wave height | | | |
|---|---|---|---|---|---|---|---|---|
| | Frequency of K-S test passed | Percentage of K-S test passed (%) | Frequency of optimal function | Percentage of optimal function (%) | Frequency of K-S test passed | Percentage of K-S test passed (%) | Frequency of optimal function | Percentage of optimal function (%) |
| Gamma | 1508 | 90.57 | 183 | 10.99 | 1464 | 87.93 | 159 | 9.55 |
| Exponential | 1567 | 94.11 | 216 | 12.97 | 1076 | 64.62 | 95 | 5.71 |

| | | | | | | | | |
|---|---|---|---|---|---|---|---|---|
| **Gumbel (right)** | 1615 | 97.00 | 350 | 21.02 | 1629 | 97.84 | 149 | 8.95 |
| **Weibull (max)** | 1469 | 88.23 | 416 | 24.98 | 300 | 18.02 | 494 | 29.66 |
| **GEV** | **1665** | **100.00** | **500** | **30.04** | **1657** | **99.52** | **768** | **46.13** |

**16. Figure3: What about complementing these figures with confidence/prediction intervals?**

**Response:** Thanks for your suggestion. We add 95% confidence intervals to Figure 3a and Figure 3c in the revised manuscript.

[Figure]

**Figure 3: Fitting results of the PDF and CDF of the SH and SWH based on the GEV function (using node (110.5142° E, 20.2768° N) as an example)**

**17. L249: "*In general, the SWH increases with an increasing return period ...*" In general? Distributions are monotonic functions: higher quantiles always correspond to higher return period for whatever (continuous) random variable... always, not "in general"!**

**Response:** We agree that "in general" does not make sense here, and the sentence is deleted.

18. **Fig. 6: This figure (along with Fig. 7) is uninformative. The shape of the surfaces of generic joint PDFs and CDFs is well known. What matters is some diagnostic plot showing the goodness of fit (with uncertainty!)**

    **Response:** We have removed the two example figures, Fig. 6 and Fig. 7, from the manuscript. In addition, we calculated the goodness of fit for each grid, filtered the optimal copula function based on this, and did not show the diagnostic plots separately anymore.

19. **L270: "*When the intensity values of the two disaster-causing factors are equivalent,* $RP_\cap$ *is greater than* $RP_\cup$*, which indicates that* $P_\cap$ *is smaller than* $P_\cup$*."* This is not a result but the effect of theoretical constraints (see here https://link.springer.com/article/10.1007/s00477-014-0916-1)**

    **Response:** Thanks for your comments. According to Eq. 2 - Eq. 5, when the intensity values of the two hazards are equivalent, $RP_\cap > RP_\cup$ and $P_\cap < P_\cup$ is evident. So, it is removed in the revised manuscript.

    **Line 260-261:** Based on the optimal marginal distribution function and copula function, we calculate $RP_\cap$, $RP_\cup$, and $RP_|$ of SHs and SWHs.

20. **Figs 8-10: These figures just report what is expected according to the monotonic nature of bivariate distributions.**

    **Response:** Thanks for your comments, but we believe that Fig. 8 - Fig. 10 can provide valuable info to readers. Even theoretically, the distributions are monotonic. So no revision is made.

    On the one hand, they show the spatial distribution of simultaneous, joint, and conditional probabilities for different combinations of return periods for storm surges and waves, showing the high and low probability of occurrence in each region and thus highlighting regional differences.

    On the other hand, they also express the characteristic monotonically increasing (decreasing) change in the simultaneous, joint, and conditional probability as the return period of one variable is held constant and as the return period of the other variable increases, laying the groundwork for later research on reducing the bivariate probability by increasing the univariate return period protection criteria.

21. **L315: "$P_\&$" was not defined in the text. I guess it refers to the discretised classes in Sect. 3.3.3 and Fig. 11.**

    **Response:** Thanks for your reminder. The $P_\&$ in this text refers to the bivariate probability of the discrete hazard class, which is calculated as Eq. 8 in Sect. 3.3.3, and we have added $P_\&$ to Eq. 8 in the revised manuscript. In addition, $P_\cup$ in L288 of the original manuscript is incorrect and should be changed to $P_\&$.

    **Line 178-179:** We calculate the bivariate probabilities for discretized hazard level combination scenarios based on the marginal and copula functions of the storm surge and

wave. The calculation formula is as follows:

$$P_\& = P(x_1 < X \le x_2, y_1 < Y \le y_2)$$
$$= P(X \le x_2, Y \le y_2) - P(X \le x_2, Y \le y_1) - P(X \le x_1, Y \le y_2) + P(X \le x_1, Y \quad (8)$$
$$\le y_1) = F_{X,Y}(x_2, y_2) - F_{X,Y}(x_2, y_1) - F_{X,Y}(x_1, y_2) + F_{X,Y}(x_1, y_1)$$

**Line 288-290:** We calculated two-dimensional $P_\&$ based on Eq. 8 for all nodes with discretized combinations of SH and SWH for a total of 25 scenarios and interpolate them into 1 km raster data using the cubic spline interpolation method (Figure 9).

22. **L409-410: These statements are not related to the preceding text; they are just generic sentences.**

**Response:** Thanks for the comments. The sentence was supposed to explain that only with joint probability available is not enough for design criteria determination. A method to determine the specific values of SH and SWH is needed. Please refer to comment #4. So, we reorganized the text as follows.

**Line 394-397:** Since the joint probability distribution of bivariate is a 3-dimensional surface. In order to obtain the specific scaler values of the two hazards as design criteria, in this study, the method for estimating the minimum return periods of SH and SWH was implemented, given their estimated joint probability distribution as a constraint.

---

## Author Response (AR1)

**General response:** We sincerely thank the editor and all the reviewers for their valuable feedback, which we have used to improve the quality of our manuscript. The reviewer comments are provided below in **bold font,** and specific concerns have been numbered. Our responses are given in normal font, and changes/additions to the manuscript are given in blue text.

**Responses to Referee #1**

**General comments**

**I regret to say that, in my opinion, the quality of this paper is not suitable for publication. Language is poor and technical terms are associated rather randomly resulting in nonsense sentences (e.g. L69, 74, 135, 143, 154, 197, just to mention a few).**

**Leaving aside the general confusion of the text (denoting lack of familiarity with theoretical concepts), the data analysis is the usual superficial application of bivariate copulas already reported in hundred of papers, iterating widespread mistakes and lacking whatever uncertainty assessment (I cannot understand why uncertainty assessment is fundamental to provide a decent univariate frequency analysis, but it suddenly disappears and is neglected when moving to multivariate frequency analysis, which is affected by the additional uncertainty related to the unknown dependence structure).**

**Some results that are interpreted as empirical findings are just theoretical constraints that do not provide any insight.**

**Justifying a paper with sentences like "*Presently, a few scholars have conducted studies on the interaction and joint distribution of storm surges and waves*" makes no sense: the use of joint distributions and copulas have been used in coastal/ocean engineering for at least 15 years worldwide to model not only surge and significant wave height but also other met-ocean variables; these methods have also been incorporated in national guidelines (e.g. UK and Netherlands, to mention a few) for several years, and are subject to ongoing improvement and update. So, please perform a decent preliminary literature review before running out-of-the-shelf computer packages.**

> **Response:** Your comments and suggestions are appreciated. We believe most of them are valuable to the improvement of this manuscript, though we are unable to agree with all of them. Kindly take your time to find our detailed response and revisions below.

**Specific comments**

1. **L17: "*the surge height shows an increasing trend closer to the coastline*" ?? -> the surge height shows higher values closer to the coastline.**

   **Response:** Thanks for the suggestion on language improvement.

   **Line 20-21:** Additionally, SH shows higher values as locations get closer to the coastline, and SWH becomes higher further from the coastline.

2. **L18:** *"when one variable is constant, the simultaneous, joint, and conditional risk probability tends to decrease as the other variable increases."* **Marginal and joint distributions are monotonic functions! This behaviour is related to their mathematical properties and has nothing to do with the analysed variables.**

**Response:** The marginal and bivariate cumulative distribution is monotone increasing functions. Correspondingly, the simultaneous, joint, and conditional probabilities decrease monotonically. As this is a mathematical property of the function, we have removed this sentence from the abstract.

3. **L58:** *"the study of the joint probability distribution of tropical cyclone storm surges and waves is conducive to improving the accuracy and precision of joint hazard assessment."* **Yes, this is true if we neglect the fact that joint distributions are affected by the same uncertainty of their marginals and the additional uncertainty of the dependence structure.**

**Response:** Uncertainty is of great importance when we quantify their joint probability. But even with the existence of uncertainty or even greater uncertainty, which no one denies, joint probability analysis can still provide valuable insights. The text was revised as below, with uncertainty mentioned.

**Line 38-45:** However, strong storm surges and waves often occur simultaneously during tropical cyclone events, which often cause greater impact than estimated only by a single variate due to the cascading effects of multi-hazards. For example, when high waves near the coast take place along strong storm surges, the overtopping and overflowing at sea dyke can lead to a large area of inundation and severe damage to coastal facilities (Rao et al., 2012; Hughes and Nadal, 2009; Pan et al., 2019). Similarly, rising sea levels due to storm surges would improve the probability of wave overtopping (Pan et al., 2013; Li et al., 2012). The concurrent interaction between storm surges and waves often assesses the effects of multi-hazards with significant uncertainties. Some studies have investigated the physical interaction of storm surges and waves through numerical simulation by coupling storm surge and wave models (Xie et al., 2016; Kimf et al., 2016; Brown, 2010) for specific events.

4. **L69:** *"However, the constructed joint distribution is still a probabilistic result, and further search for constraint relations is needed to provide a basis and guidance for disaster prevention and mitigation. Therefore, this paper quantitatively analyzes the occurrence probability of storm surge and wave combinations based on the fitting results of the copula function."* **Words have a meaning and should be used accordingly! First, you say that "constraint relations" are needed because joint distributions are still a probabilistic (which seems to be a minus, whatever that sentence means), and then joint distributions are used neglecting constraints. Those sentences contradict each other.**

**Response:** We reorganized the text as follows, expressing that, even with a joint probability distribution (a 3d surface or a 2d curve with the probability of one variate fixed) available, it is not enough for setting design criteria. And what needs to do furtherly is to find a specific combination of surge and wave probability, and obtain their respective SH and

SWH (i.e., two scalar values).

Please also refer to comment # 22, which was supposed to explain the same thing.

**Line 57-60:** In addition, even with the joint probability of bivariate estimation, only an intercepted curve can be obtained since their probability is a three-dimensional surface. It is not applicable for actual protection design without a specific scalar value for SH or SWH.

5. **L74: "*In the design process of sea dikes, breakwaters, and harbors, the surge height and significant wave height in different return periods is separately considered… disregarding the correlation between storm surge and so that the calculated water level may be underestimated or overestimated.*" This is incorrect, underestimation/overestimation means that an estimate is smaller/larger than a true value. In real-world applications, (i) the true value is unknown, so you never know if an estimator overestimates/underestimates, and (ii) univariate probabilities cannot be compared with joint probabilities as they refer to different processes! The choice of the required probabilistic model depends on the specific problem/failure mechanism. Univariate analysis is perfectly fine if there is a unique target design variable, and it correctly estimates the required probability (these concepts are explained here https://link.springer.com/article/10.1007/s00477-014-0916-1).**

**Response:** (1) For sure, it is not easy or even practically impossible to estimate the so-called TRUE value of the intensity for a given probability. But it is also fair and safe to say that, without considering their interaction, it is highly possible that the TRUE value will be underestimated or overestimated, since the physical interaction mechanism between surge and wave is there. (2) On the other hand, the value of univariate analysis and its application of course are there, which is not the focus of this manuscript. That said, we only want to emphasize the importance of joint probability in the text. But we did revise the sentence a little bit to avoid misunderstanding.

**Line 38-45:** However, strong storm surges and waves often occur simultaneously during tropical cyclone events, which often cause greater impact than estimated only by a single variate due to the cascading effects of multi-hazards. For example, when high waves near the coast take place along strong storm surges, the overtopping and overflowing at sea dyke can lead to a large area of inundation and severe damage to coastal facilities (Rao et al., 2012; Hughes and Nadal, 2009; Pan et al., 2019). Similarly, rising sea levels due to storm surges would improve the probability of wave overtopping (Pan et al., 2013; Li et al., 2012). The concurrent interaction between storm surges and waves often assesses the effects of multi-hazards with significant uncertainties. Some studies have investigated the physical interaction of storm surges and waves through numerical simulation by coupling storm surge and wave models (Xie et al., 2016; Kimf et al., 2016; Brown, 2010) for specific events.

6. **L80: "*optimize*" -> fit. As the confusion due to meaningless terminology already affects a significant part of the literature, please, use consistent technical terms without inventing a new vocabulary!**

**Response:** Relax, but we did not mean to create a new term, and we did not create a new term. When we use the word "optimize," we mean to select the optimal marginal

distribution function and proper copula function through the AIC, BIC, and goodness of fit of the K-S test for all nodes, not just a simple fitting process. In order to clarify that, the text was revised as follows.

**Line 61-65:** In this study, we aim to explore the joint probability characteristics of tropical cyclone storm surges and waves for large coastal areas and to investigate the methods and steps for selecting the protection standard of sea dikes. Firstly, we fit the marginal and copula function of nodes in the study area based on the numerically simulated tropical cyclone SH and SWH from 1949 to 2013. Next, the optimal functions are selected for every modeling node based on the Kolmogorov-Smirnov (K-S) test, AIC, and BIC.

7. **L81: *"function by the passing rate of the K-S test."* When performing tests, the only meaningful "rate" is the rejection rate! (see here https://www.sciencedirect.com/science/article/pii/S0309170817305845 for an explanation)**

**Response:** Thanks for your comment and suggestion. The term "passing rate" was misused in the original text, and it did not refer to the results of the K-S test, and we have changed the original statement in the revised manuscript. In the process of selecting the optimal marginal distribution function and copula function, the goodness-of-fit of each marginal distribution function and copula function is calculated by the K-S test for each node. If the p-value of the K-S test of a node is greater than 0.05, the hypothesis of "the sample obeys a certain theoretical distribution" is not rejected, which means that the node passes the K-S test. Then, the ratio of the number of nodes passing the K-S test to the number of all nodes is calculated, and based on this, the optimal function is determined by combining the AIC, BIC, and D-values of the K-S tests.

**Line 62-65:** Firstly, we fit the marginal and copula function of nodes in the study area based on the numerically simulated tropical cyclone SH and SWH from 1949 to 2013. Next, the optimal functions are selected for every modeling node based on the Kolmogorov-Smirnov (K-S) test, AIC, and BIC.

8. **L135: If the result of rewording concepts is this one, then it is preferable copying and pasting from some good paper/handbook. These sentences are just a set of randomly chosen words that make no sense whatsoever.**

**Response:** Thanks for the suggestion on language.

**Line 119-124:** Sklar (Sklar, 1973) elucidates the role that copula play in the relationship between multivariate distribution and their univariate margins distribution, and states that any multivariate joint distribution can be described by a univariate marginal distribution function and a couple describing the dependence structure between the variables (Nelsen, 2006). Let $F(x)$ and $G(y)$ be the marginal distributions of $x$ and $y$, $C$ is the copula, and $H(x,y) = C(F(x), G(y))$, where $H$ is the bivariate joint distribution function of $x$ and $y$ (Serinaldi, 2015). Therefore, the copula function is widely utilized in multi-hazard joint probability analysis of natural disasters (Chen et al., 2019; Lee et al., 2013).

9. **L151: Archimedean, elliptical and quadratic are not families but classes of copulas,**

**and they are not the only existing classes. Archimedean copulas can be multiparametric as well! Please, do not use models without knowing the underlying theory, as the result is just a collection of misconceptions, meaningless statements, and misinterpreted figures and tables.**

**Response:** We believe this is a language problem, which has been revised by rewording. Joint probability analysis is not something like unsolved mathematical puzzles. And we believe avoiding subjective judgment with no consolidated evidence will improve the quality of the review.

**Line 138-141:** There are a variety of copulas families, including Meta-elliptical copulas (normal and t), Archimedean copulas (Clayton, Gumbel, Frank, and Ali-Mikhail-Haq), Extreme Value copulas (Gumbel, Husler-Reiss, Galambos, Tawn, and t-EV), and the other families (Plackett and Farlie-Gumbel-Morgenstern) (Chen and Guo, 2019). Among these copulas, the Archimedean copula is more popular for hydrologic applications.

10. **L154: *"analyze the joint probabilities of the marginal distributions of two variables"* -> "the joint probabilities of two variables". Joint probabilities of the marginal distributions do not exist!**

**Response:** Thanks for the comments. Revision is made as follows.

**Line 141-143:** The commonly employed Archimedean copula functions include Gumbel, Clayton, and Frank (Table 1), which are selected to analyze the joint probabilities of two variables, the SHs, and SWHs of a tropical cyclone. Then the maximum likelihood method is used to estimate the parameters of the copula function.

11. **L175: The conditional probabilities and the corresponding conditional distributions and return periods are not joint (multidimensional) but univariate (unidimensional).**

**Response:** Thanks for your comment. But we do not think your comment is correct, especially in the context of this manuscript.

We believe that the conditional probabilities and return periods can be bivariate distributions.

The concept of the conditional probability of a multidimensional random variable is the probability of $(X > x)$ occurring under the condition that $(Y > y)$, which is calculated by the formula $P_| = P\big((X > x)|(Y > y)\big) = \frac{P(X>x,Y>y)}{P(Y>y)}$. Therefore, it is a conditional distribution of the two-dimensional variable $(X, Y)$. In addition, a preliminary introduction to the concepts of multivariate probabilities and return periods is provided in the paper (https://link.springer.com/article/10.1007/s00477-014-0916-1), which also states that conditional probabilities and conditional return periods are multivariate.

12. **L197: *"The marginal and joint probabilities of storm surge and wave scenarios cannot be directly employed as reference values for engineering protection standards."* Return periods are just indicators derived from probabilities: if the former can be used, the latter can be used as well.**

**Response:** According to your suggestion, we have deleted "marginal" from the original manuscript and only proposed that the joint probability cannot be used as a reference value.

This is mainly because the intensity values of storm surges and waves under different univariate return periods can be used as a reference for marine engineering protection. However, the bivariate joint probability/return period is a three-dimensional surface. The intercepted curve under the specified occurrence probability or return period is a curve, so it is difficult to be used directly as a reference value for engineering protection criteria.

**Line 183-185:** The joint probabilities of storm surge and wave scenarios are hard to be directly employed as reference values for engineering protection standards, since the bivariate joint probability and joint return period is a three-dimensional surface, and the intercepted curve under the specified occurrence probability or return period is a curve, not a scalar.

13. **Sect. 3.4.2: this awkward method is quite useless because AND and OR return level curves are complementary, and their surfaces are monotonic and diagonally symmetric for the considered Archimedean copulas. This means that the point estimate (x, y) resulting from the (unnecessary) non-linear optimization is just the intersection of the main diagonal of the copula and the OR level curve with RP = k. No optimization is required, just some familiarity with the theoretical properties of the models one intends to use.**

**Response:** Thanks for your comments. But we have a different opinion from you, and no revision has been made.

We believe it is essential to use a non-linear optimization method to estimate optimal design values for storm surge and wave combinations. This is because although $RP_\cap$ and $RP_\cup$ are complementary and the three-dimensional surface is monotonic, they are not perfectly diagonally symmetric. Therefore it is not the intersection of the main diagonal and $RP_\cup = K$ that corresponds to the largest value of $RP_\cap$.

14. **L214: "*determine whether it passes the 95% significance level.*" "Passing the 95% significance level" makes not sense. A test can only reject or not reject the reference/null hypothesis at a given significance level. The significance level is not 95% but 5%.**

**Response:** Thanks for your comments. We have now changed the wording to "*the p-value of K-S test is used to determine whether the hypothesis that the sample obeys a certain theoretical distribution is rejected*" in the revised manuscript.

**Line 201-204:** Firstly, the time series of the bivariate annual maximum value for all nodes are fitted with five marginal functions, including Gumbel, Weibull, gamma, exponential, and generalized extreme value (GEV). Next, the p-value of the K-S test is used to determine whether the hypothesis that the sample obeys a certain theoretical distribution is rejected.

15. **Table 3 and corresponding discussion: Using the rate of no rejection to select the best model makes no sense for several reasons:**

**1) "No rejection" does not mean acceptance, and if two or more models are not rejected for a given data set, the only possible conclusion is that the information/data is not enough to discriminate among the models.**

**2) The comparison includes distributions with different number of parameters (1-parameter Exponential, 2-parameter Gamma, Gumbel and Weibull (assuming that the Weibull does not include a location parameter), and the 3-parameter GEV). So, GEV has the obvious advantage of being more flexible due to higher parameterization.**

**3) Performing a test on 1665 nodes (i.e., 1665 times) is a multiple testing exercise. Under independence (which is not valid in this case), if the null hypothesis is valid, the number of no rejections has an expected value equal to 1582 (95%), and a binomial distribution, thus meaning that the 95% confidence interval of the number of no rejections is (1564, 1599). Therefore, Exponential and Gumbel cannot be discarded for the variable SH.**

**However, met-ocean variables come from a model and refer to grid points of a connected area, they are surely strongly correlated in space. This means that the binomial distribution strongly underestimates the actual variability of the number of no rejections because of information redundancy. Therefore, other distributions for both SH and SWH might not be discarded. This is not surprising as these distributions are fitted to just 60/65 annual maxima, i.e. a very small sample size.**

**4) "No rejection" rate for Weibull and SWH is reasonably wrong because (i) WH/SWH (in deep water) has been historically modelled by Rayleigh distribution, which a particular case of Weibull, (ii) Weibull was shown to be a very good model in shallow water for WH (see https://doi.org/10.1016/j.coastaleng.2022.104130), and (iii) Weibull and 3-paremeter Weibull are closely related to GEV, and Weibull is also a pre-asymptotic distribution in EVT.**

**In summary the selection strategy seems to work just because it neglects the foregoing theoretical aspects.**

**Response:** Thanks for pointing out the problems in the marginal distribution selection process. We changed/refined the selection process as follows:

(1) The p-value of the K-S test is used to determine whether each node rejects the hypothesis that the sample obeys a certain theoretical distribution (Table 3).

(2) The optimal function is screened by three metrics, AIC, BIC, and the D-value of the K-S test. The smaller the AIC, BIC, and D-value of the K-S test, the better the fit, thus determining the optimal marginal function for each node.

(3) The percentage of each optimal function to the number of all nodes was calculated (Table 3), and the function with the highest rate was selected as the optimal marginal function for the storm surges and waves.

Through the above steps, GEV was finally determined as the marginal function for storm surges and waves:

(1) GEV belongs to 3 parameters, and the parameter fitting is relatively flexible.

(2) The no-rejection numbers of the fitted GEV functions for storm surge and waves are 1665 and 1657, respectively, both of which are in the 95% confidence interval.

(4) Among all functions, the GEV function has the highest percentage of preferences, 30% and 27%, respectively.

**Line 131-135:** In this study, the maximum likelihood method is used to estimate the function parameters, based on which the optimal marginal functions for SHs and SWHs are screened by the following steps: Firstly, the p-value of the K-S test is used to determine whether each node rejects the hypothesis that the samples obey a certain functional distribution. Secondly, the optimal function for each node is screened by the three metrics, AIC, BIC, and D-value of the K-S test. The smaller the AIC, BIC, and D-value of the K-S test, the better the goodness of fit, thus determining the optimal marginal function for each node.

**Line 204-206:** Then, we count the number of nodes passing the K-S test for each function and their percentage of all nodes. Finally, the number of nodes and their percentage of each function being selected as optimal is calculated according to the steps for optimal function selection in Section 3.1 (Table 1).

**Line 212-215:** Based on the statistical results, it is found that for fitting the SH, the K-S test of the GEV function had the highest no-rejection rate of 100%, and the corresponding optimal ratio was 30.04%, so GEV is set as the optimal marginal function in this study. For SWH fitting, the number of nodes with no rejection in the K-S test of the GEV function is 1657, accounting for 99.52% of the total number of nodes, and the corresponding percentage of preferences is also higher than that of other functions.

Table 1 Frequency and percentage of five functions passing the K-S test and the optimal function for all nodes of SH and SWH

| Marginal function | Surge height | | | | Significant wave height | | | |
|---|---|---|---|---|---|---|---|---|
| | Frequency of K-S test passed | Percentage of K-S test passed (%) | Frequency of optimal function | Percentage of optimal function (%) | Frequency of K-S test passed | Percentage of K-S test passed (%) | Frequency of optimal function | Percentage of optimal function (%) |
| Gamma | 1508 | 90.57 | 183 | 10.99 | 1464 | 87.93 | 159 | 9.55 |
| Exponential | 1567 | 94.11 | 216 | 12.97 | 1076 | 64.62 | 95 | 5.71 |
| Gumbel (right) | 1615 | 97.00 | 350 | 21.02 | 1629 | 97.84 | 149 | 8.95 |
| Weibull (max) | 1469 | 88.23 | 416 | 24.98 | 300 | 18.02 | 494 | 29.66 |
| GEV | 1665 | 100.00 | 500 | 30.04 | 1657 | 99.52 | 768 | 46.13 |

**16. Figure3: What about complementing these figures with confidence/prediction intervals?**

**Response:** Thanks for your suggestion. We add 95% confidence intervals to Figure 3a and

Figure 3c in the revised manuscript.

[Figure]

**Figure 3: Fitting results of the PDF and CDF of the SH and SWH based on the GEV function (using node (110.5142° E, 20.2768° N) as an example)**

17. **L249: "*In general, the SWH increases with an increasing return period ...*" In general? Distributions are monotonic functions: higher quantiles always correspond to higher return period for whatever (continuous) random variable… always, not "in general"!**

    **Response:** We agree that "in general" does not make sense here, and the sentence is deleted.

18. **Fig. 6: This figure (along with Fig. 7) is uninformative. The shape of the surfaces of generic joint PDFs and CDFs is well known. What matters is some diagnostic plot showing the goodness of fit (with uncertainty!)**

    **Response:** We have removed the two example figures, Fig. 6 and Fig. 7, from the manuscript. In addition, we calculated the goodness of fit for each grid, filtered the optimal copula function based on this, and did not show the diagnostic plots separately anymore.

19. **L270: "*When the intensity values of the two disaster-causing factors are equivalent, $RP_\cap$ is greater than $RP_\cup$, which indicates that $P_\cap$ is smaller than $P_\cup$.*" This is not a result but the effect of theoretical constraints (see here https://link.springer.com/article/10.1007/s00477-014-0916-1)**

**Response:** Thanks for your comments. According to Eq. 2 - Eq. 5, when the intensity values of the two disaster-causing factors are equivalent, $RP_\cap > RP_\cup$ and $P_\cap < P_\cup$ is evident. So, it is removed in the revised manuscript.

**Line 256:** Based on the optimal marginal function and copula function, we calculate $RP_\cap$, $RP_\cup$, and $RP_|$ of SHs and SWHs.

20. **Figs 8-10: These figures just report what is expected according to the monotonic nature of bivariate distributions.**

**Response:** Thanks for your comments, but we believe that Fig. 8 - Fig. 10 can provide valuable info to readers. Even theoretically, the distributions are monotonic. So no revision is made.

On the one hand, they show the spatial distribution of simultaneous, joint, and conditional probabilities for different combinations of return periods for storm surges and waves, showing the high and low probability of occurrence in each region and thus highlighting regional differences.

On the other hand, they also express the characteristic monotonically increasing (decreasing) change in the simultaneous, joint, and conditional probability as the return period of one variable is held constant and as the return period of the other variable increases, laying the groundwork for later research on reducing the bivariate probability by increasing the univariate return period protection criteria.

21. **L315: "$P_\&$" was not defined in the text. I guess it refers to the discretised classes in Sect. 3.3.3 and Fig. 11.**

**Response:** Thanks for your reminder. The $P_\&$ in this text refers to the bivariate probability of the discrete hazard class, which is calculated as Eq. 8 in Sect. 3.3.3, and we have added $P_\&$ to Eq. 8 in the revised manuscript. In addition, $P_\cup$ in L288 of the original manuscript is incorrect and should be changed to $P_\&$.

**Line 170-172:** We calculate the bivariate probabilities for discretized hazard level combination scenarios based on the marginal and copula functions of the storm surge and wave. The formula is as follows:

$$P_\& = P(x_1 < X \leq x_2, y_1 < Y \leq y_2)$$
$$= P(X \leq x_2, Y \leq y_2) - P(X \leq x_2, Y \leq y_1) - P(X \leq x_1, Y \leq y_2) + P(X \leq x_1, Y \leq y_1) = F_{X,Y}(x_2, y_2) - F_{X,Y}(x_2, y_1) - F_{X,Y}(x_1, y_2) + F_{X,Y}(x_1, y_1) \tag{8}$$

**Line 285-287:** We calculate the combined scenario probability $P_\&$ based on Eq. 8 for all nodes with different combinations of SH and SWH for a total of 25 scenarios and interpolate them into 1 km raster data using the cubic spline interpolation method (Figure 9).

**22. L409-410: These statements are not related to the preceding text; they are just generic sentences.**

**Response:** Thanks for the comments. The sentence was supposed to explain that only with joint probability available is not enough for design criteria determination. A method to determine the specific values of SH and SWH is needed. Please refer to comment #4. So, we reorganized the text as follows.

**Line 383-386:** Therefore, the development of appropriate design surge heights and significant wave heights can be effective in reducing hazard impacts, which allows coastal areas to cope with a high bivariate probability. To obtain the specific scaler values of the two hazards as design criteria, in this study, the method for estimating the minimum return periods of SH and SWH was implemented, given their estimated joint probability distribution as a constraint.

**Responses to Referee #2**

**General comments**

The manuscript describes the analysis of the joint probability of storm surge and the significant wave height caused by tropical cyclones. The analysis was applied to a specific case study (Eastern coast of the Leizhou Penisula and Hainan Island of China).

Although the approach can be interesting, the paper presents several shortcomings related both to the methodology and to the presentation of results. The approach adopted for the description of the analysis is not very detailed. In particular, the description of the numerical models does not allow for an adequate understanding of their performance and the reliability of the results. More details about the setup of the numerical models and the validation result should be provided. Moreover, there is a lack of physical explanation of the phenomena simulated and related results.

Overall, the paper, although interesting, cannot be published in its present form.

**Response:** We greatly appreciate your suggestions and comments in reviewing the manuscript. Please kindly find the detailed responses and revisons to your comments below.

**Specific comments**

**23. How many tropical cyclones were used in this study? (line 95: 87, line 109: 119, line 119: 102).**

**Response:** Thanks for your comments. As the simulation results of the tropical cyclone storm surge and wave in this manuscript were obtained from different institutions, the number of simulations for historical tropical cyclone storm surges and waves was different, with a total of 119 events for storm surges and 102 events for waves. In this study, 86 tropical cyclone storm surges and waves that simultaneously affected the study area from 1949 to 2013 were finally screened for joint probability analysis. To reduce misunderstanding, we have set the number of TC events to 86 in the revised manuscript.

**Line 81-83:** The TC surge heights (SHs) dataset is obtained from the Ocean University of China, mainly through the ADvanced CIRCulation model (ADCIRC) simulations, which includes the SHs of 86 TCs affecting the eastern coast of the Leizhou Peninsula and Hainan Island from 1949 to 2013 (Liu et al., 2018; Li et al., 2016).

**Line 95-97:** The TC significant wave heights (SWHs) dataset is also obtained from the Ocean University of China, mainly through the Simulating WAves Nearshore (SWAN) model, and includes the SWHs of 86 TC events affecting the study area from 1949 to 2013 (Li et al., 2016).

**Line 115-117:** Based on the dataset of surge height (SH) and significant wave height (SWH) of tropical cyclones, we screen 86 historical tropical cyclones (TC) events that simultaneously affected the study area from 1949 to 2013 for joint probability

characteristics analysis of storm surge and wave.

24. **To understand the result of the numerical models, the authors should provide a depth map of the study area.**

    **Response:** Thanks for your suggestions. We agree that a depth map is needed. So we add a sentence to show the source of a depth map of the study area (Liu et al., 2018). In this study, we focus more on the joint probability analysis using the simulation results of tropical cyclone storm surges and waves, where the storm surge and wave datasets are derived from the model simulation results of the Ocean University of China, and the corresponding references (Liu et al., 2018; Li et al., 2016) are cited in the manuscript, and the source of the data is indicated and acknowledged in the acknowledgments.

    **Line 83-86:** The TC surge heights (SHs) dataset is obtained from the Ocean University of China, mainly through the ADvanced CIRCulation model (ADCIRC) simulations, which includes the SHs of 86 TCs affecting the eastern coast of the Leizhou Peninsula and Hainan Island from 1949 to 2013 (Liu et al., 2018; Li et al., 2016). The previous study provides a water depth map for the study area (Liu et al., 2018).

    **Line 97-99:** The TC significant wave heights (SWHs) dataset is also obtained from the Ocean University of China, mainly through the Simulating WAves Nearshore (SWAN) model, and includes the SWHs of 86 TC events affecting the study area from 1949 to 2013 (Li et al., 2016).

    **Line 399-401:** We are grateful to Xing Liu of the Ocean University of China for providing the simulation data of storm surges and waves for historical tropical cyclone events.

    *Liu, X., Jiang, W., Yang, B., and Baugh, J.: Numerical study on factors influencing typhoon-induced storm surge distribution in Zhanjiang Harbor, Estuar. Coast. Shelf Sci., 215, 39–51, https://doi.org/10.1016/j.ecss.2018.09.019, 2018.*

    *Li, J., Fang, W., Zhang, X., Cao, S., Yang, X., Liu, X., and Sun, J.: Similar tropical cyclone retrieval method for rapid potential storm surge and wave disaster loss assessment based on multiple hazard indictors, Mar. Sci., 40, 49–60, https://doi.org/10.11759/hykx20151104001, 2016.*

25. **The authors do not provide the criteria adopted for the generation of the unstructured grid. Were there any convergence tests?**

    **Response:** SMS software is used in the model to construct a refined and unstructured grid of offshore of the eastern Leizhou Peninsula. A description of the unstructured grid can be found in the literature (Li et al., 2016), as follows:

    A nesting technique is used in this study to distinguish the local atmospheric forcing from the remote atmospheric forcing. By setting an open boundary for the subdomain, we were able to force the subdomain either with or without remote atmospheric forcing. This can also balance the demands of high resolution in ZH and the associated computation costs. As shown in Fig. 1, the full domain covers most of the South China Sea, and the subdomain only occupies a small area surrounding ZH. There are 36367 elements and 18606 nodes in

the full domain with the resolution varying from 0.7 to 20 km. In the subdomain, there are 27487 elements and 14607 nodes with the resolution varying from 0.18 to 3 km. The time step used in the model is 3 s.

26. **How were the boundary conditions of the numerical models set?**

**Response:** Thanks for your question. The following is a description of the boundary conditions in the literature (Li et al., 2016).

The full domain is driven by atmospheric forcing at the surface and surge elevation inversed from the sea surface atmospheric pressure at the open boundary. The boundary condition to force the surge in the subdomain is the time series of the water level on each boundary nodes, which includes both the tide elevation of 8 major constituents (M2, S2, N2, K2, K1, O1, P1, and Q1) in that area from OSU Tidal Prediction Software (Egbert and Erofeeva, 2002) and the surge elevation extracted from the full domain results. A tide model is also run in the subdomain with only the tidal signal including the same 8 constituents being applied at the open boundary.

27. **Which are the governing equations of the two models? Why were these models selected?**

**Response:** The wave model has entered a mature stage of development. The third-generation wave model SWAN has the advantage of high computational accuracy and stability and has been widely used in numerical simulations of offshore waters. We added related info in the revised manuscript.

In addition, the ADCIRC model integrates the effects of various boundary conditions and external forcing, uses triangular grids with different resolutions, making it more computationally efficient and well applicable in numerical simulations, and is widely used in areas such as marine, coastal and small-scale estuary systems.

Therefore, the SWAN and ADCIRC model simulations of tropical cyclone waves and storm surges were chosen for the joint probabilistic analysis.

The governing equations of the SWAN model are as follows:

$$\frac{\partial N}{\partial t} + \frac{\partial}{\partial \lambda}[(c_\lambda + U)N] + cos^{-1}\varphi \frac{\partial}{\partial \varphi}[(c_\varphi + V)N\ cos\ \varphi] + \frac{\partial}{\partial \theta}[c_\theta N] + \frac{\partial}{\partial \sigma}[c_\sigma N] = \frac{S_{tot}}{\sigma}$$

The wave action density $N(t, \lambda, \varphi, \sigma, \theta)$ is allowed to evolve in time $(t)$, geographic space $(\lambda, \varphi)$ and spectral space (with relative frequencies $\sigma$ and directions $\theta$), $(c_\lambda, c_\varphi)$ is the group velocity, $(U, V)$ is the ambient current, and $c_\theta$ and $c_\sigma$ are the propagation velocities in the $\theta$- and $\sigma$- spaces, the source terms $S_{tot}$ represent wave growth by wind.

ADCIRC computes water levels via the solution of the Generalized Wave Continuity Equation (GWCE), which is a combined and differentiated form of the continuity and momentum equations:

$$\frac{\partial^2 \zeta}{\partial t^2} + \tau_0 \frac{\partial \zeta}{\partial t} + Sp\frac{\partial \widetilde{J_\lambda}}{\partial \lambda} + \frac{\partial \widetilde{J_\varphi}}{\partial \varphi} - SpUH\frac{\partial \tau_0}{\partial \lambda} - VH\frac{\partial \tau_0}{\partial \varphi} = 0$$

and the currents are obtained from the vertically-integrated momentum equations:

$$\frac{\partial U}{\partial t} + S_p U \frac{\partial U}{\partial \lambda} + V \frac{\partial U}{\partial \varphi} - fV = -gS_p \frac{\partial}{\partial \lambda}\left[\zeta + \frac{P_s}{g\rho_0} - \alpha\eta\right] + \frac{\tau_{s\lambda,winds} + \tau_{s\lambda,winds} - \tau_{b\lambda}}{\rho_0 H} + \frac{M_\lambda - D_\lambda}{H}$$

$$\frac{\partial V}{\partial t} + S_p U \frac{\partial V}{\partial \lambda} + V \frac{\partial V}{\partial \varphi} - fU = -g \frac{\partial}{\partial \varphi}\left[\zeta + \frac{P_s}{g\rho_0} - \alpha\eta\right] + \frac{\tau_{s\varphi,winds} + \tau_{s\varphi,winds} - \tau_{b\varphi}}{\rho_0 H} + \frac{M_\varphi - D_\varphi}{H}$$

where $H = \zeta + h$ is total water depth; $\zeta$ is the deviation of the water surface from the mean; h is bathymetric depth; $Sp = cos\varphi_0/cos\varphi$ is a spherical coordinate conversion factor and $\varphi_0$ is a reference latitude; $U$ and $V$ are depth-integrated currents in the $x-$ and $y-$ directions, respectively; $Q_\lambda = UH$ and $Q_\varphi = VH$ are fluxes per unit width; $f$ is the Coriolis parameter; $g$ is gravitational acceleration; $Ps$ is atmospheric pressure at the surface; $\rho_0$ is the reference density of water; $\eta$ is the Newtonian equilibrium tidal potential, and $\alpha$ is the effective earth elasticity factor; $\tau_{s,winds}$ and $\tau_{s,waves}$ are surface stresses due to winds and waves, respectively; $\tau_b$ is bottom stress; $M$ are lateral stress gradients; $D$ are momentum dispersion terms; and $\tau_0$ is a numerical parameter that optimizes the phase propagation properties.

**Line 97-98:** The SWAN model has the advantage of high computational accuracy and stability and has been widely used in numerical simulations of offshore waters.

**Line 84-86:** The ADCIRC model integrates the effects of various boundary conditions and external forcing and uses triangular grids with different resolutions, making it more computationally efficient and applicable in numerical simulations.

28. **Provide more details about the validation of the numerical results (station location, comparison of the storm surge and the significant wave height, performance parameters, etc.).**

**Response:** Thanks for your suggestions. As the tropical cyclone storm surge and wave datasets used in this study are obtained from the Ocean University of China. Following your suggestion, the physical metrics, spatial and temporal resolution, and model simulation accuracy are briefly described in this manuscript. As for the validation of the numerical results, the locations of the validation sites are added in Figure 1b in this manuscript, and the validation details can be found in the reference (Liu et al., 2018; Li et al., 2016).

**Line 86-91:** The simulation results are the total water level after the superposition of the water gain caused by a tropical cyclone and astronomical tide, and the time step is 30 minutes. To improve the simulation accuracy and computing speed of the hot spot area, the model adopts a triangular grid with nested small- and large-area grids, and the resolutions of different area grids are set in a gradual resolution range from 0.0039° to 0.3°. Comparing the simulation values with the measured surge height at the observation sites, we discover that the absolute standard error is 47 cm, the relative standard error is 22%, and the simulation results are similar to the observed values in most cases.

**Line 99-104:** The model also uses a triangular grid with nested small- and large-area grids and gradual resolution, but the nodes' scopes and locations differ from those of the storm

surge model. Comparing the observed data of buoy stations with the simulated values reveals that the unstructured grid can well reflect the wave variation conditions in the sea. In addition, the mean absolute and root mean square errors of the simulated results of the locally encrypted unstructured triangular grid are the smallest, indicating that the data can effectively reproduce the wave distribution during tropical cyclones.

29. **Figure 4. On the East side of Hainan Island, the surge height increased considerably. Is there some physical explanation for such phenomena?**

**Response:** Thanks for your suggestion. In the revised manuscript, we have added the appropriate explanation of the spatial distribution of storm surge heights on the eastern side of Hainan Island.

**Line 229-232:** As the return period increases, SH offers varying degrees of growth in each region, and regional differences are more pronounced, with a significant area of development in southeastern Hainan Island, which may be related to the region's location on the edge of the continental shelf and high variability in seafloor topography.

30. **Figure 5: The significant wave height is very high in the offshore region. Is there some physical explanation for such phenomena? It is suggested to consider the effect of the wave breaking.**

**Response:** Thanks for your suggestion. The significant wave height in the offshore region is relatively lower than in distant areas due to wave breaking, which we have explained in the revised manuscript.

**Line 235-237:** As shown in Figure 5, the SWH near the shore is generally smaller than that in the open sea, and there is a significant decreasing trend in SWH as it gets closer to the coastline. This finding is mainly attributed to the shallow shore depth, island obstruction, wave breaking, and seabed friction attenuation.

31. **It would seem that the effect of the sea level rising due to storm surge was not considered in the numerical simulation conducted with SWAN. If this is true, in the intermedia and shallow water the results are likely to be unreliable.**

**Response:** Great thanks to the reviewer for pointing out this. It is indeed not considered when modeling the wave. We also realized that the limitation of this wave dataset would introduce uncertainty in the joint probability estimation. In the meantime, it is challenging to simulate the wave again for this study. In order to inform the readers of this fact, we explained the limitation of this dataset in the data section. And in the discussion section, the impact of this limitation is discussed.

**Line 104-105:** It shall be noted that the effect of sea level rise due to storm surge was not considered during the SWH simulation, which will influence the accuracy of SWHs, especially in intermedia and shallow water.

**Line 387-391:** Although this study provides helpful insights into joint probability analysis of storm surges and waves using Copula functions, there are several limitations that need to be addressed in future research. One limitation is the absence of water level rise caused

by storm surges in the numerical modeling of waves, which may introduce errors in the simulation of SWHs in intermediate and shallow water. In addition, exploring the contribution of other indicators, such as long-term sea level rise as environmental hazards, can further improve the accuracy of risk assessment.

**Minor point**

1. **Figure 1. Add the location of stations used for the validation of the models.**

   **Response:** Thanks for your suggestion. We added the location of the verification stations in Figure 1 (b).

   **Line 77-79:**

[Figure]

**Figure 1: Best track and MSW of 86 TCs in this study from 1949 to 2013 (a) and the study area for the joint probability analysis of storm surges and waves of TCs (b).**

2. **Triangular network -> triangular grid.**

   **Response:** Thanks for the suggestion. We have modified the phrase "triangular network" to "triangular grid" in the revised manuscript.

   **Line 112-114:** Based on the location of the nodes of the triangular grid in the storm surge (Section 2.2) and wave datasets (Section 2.3), we select the region with a dense distribution of both as the study area, and the finalized spatial range is 110°E - 113°E, 18°N - 22°N (Figure 1b).

   **Line 87-89:** To improve the simulation accuracy and computing speed of the hot spot area, the model adopts a triangular grid with nested small- and large-area grids, and the resolutions of different area grids are set in a gradual resolution range from 0.0039° to 0.3°.

   **Line 99-100:** The model also uses a triangular grid with nested small- and large-area grids and gradual resolution, but the nodes' scopes and locations differ from those of the storm surge model.

   **Line 102-104:** In addition, the mean absolute and root mean square errors of the simulated results of the locally encrypted unstructured triangular grid are the smallest, indicating that the data can effectively reproduce the wave distribution during tropical cyclones.

**Line 196-198:** Since the different densities and locations of the triangular grids in the storm surge and wave models, we use the storm surge triangular grid nodes as the benchmark and the wave node closest to each storm surge node as the wave simulation result based on the nearest neighbor method.

3. **Table 1. Add the definitions of u and v.**

   **Response:** Thank you for the suggestion. We have added the definitions of $u$ and $v$ in the revised manuscript.

   **Line 148:** Note: $u$ and $v$ are uniform (0,1) random variables (Nelsen, 2006).

   *Nelsen R B. An introduction to copulas[M]. 2nd Edition. New York, USA: Springer, 2006.*

---

## Author Response (AR2)

Dear Editor,

Thank you very much for your suggestions on our manuscript (eggsphere-2022-847). We have revised the manuscript according to the reviewers' suggestions, and proof-read the manuscript to minimize typographical, grammatical, and bibliographical errors. We prepared three documents as requested: (1) a point-to-point reviewer response document including original comments, our response, and corresponding revisions made in the manuscript, (2) a marked-up manuscript version showing all the detailed modifications in the manuscript, and (3) a revised manuscript.

We appreciate your kind help in the process of review and revision. We look forward to further updates from you.

Sincerely,
Weihua Fang
On behalf of the co-authors

**General response:** We sincerely thank the editor and all the reviewers for their valuable feedback, which we have used to improve the quality of our manuscript. The reviewer comments are provided below in **bold font,** and specific concerns have been numbered. Our responses are given in normal font, and changes/additions to the manuscript are given in blue text.

**Responses to Referee #1**

**General comments**

**Even if the theory is classic, the topic of the paper is original for three points: 1) the joint probability analysis is applied spatially on a large area; 2) a design criteria is proposed combining simultaneous probability and joint probability; 3) an analysis of change of return period protection standards is presented.**

**Response:** We greatly appreciate your comments and suggestions. Please kindly find the detailed responses and revisions below.

1. **The notations 5a, 10a, ... is not known by the reviewer. Does it mean 5 years, 10 years, ... ? It should be understood easily by the reader.**

   **Response:** Thank you for your comment, the notations 5a, 10a, ... mean 5 years, 10 years,.... respectively. Following your suggestion, we have modified the 5a, 10a to 5-year, and 10-year return periods in the revised manuscript.

   **Line 259-260:** Based on the univariate return period formula (Eq. 5), the SH and SWH are estimated for six typical return periods of 5-year, 10-year, 20-year, 50-year, 100-year, and 200-year at all nodes.

   **Line 293-295:** In addition, based on the formula of bivariate probability (Eq. 6 and Eq. 8), $P_\cap$ and $P_\cup$ of SH and SWH are calculated for all nodes with a combination of 10-year, 20-year, 50-year, and 100-year return periods.

   **Line 299-300:** $P_\cap$ is greatest when the return period of SH and SWH is 10-year, which is higher than 0.05. $P_\cap$ is the smallest for SH and SWH of 100-year return period, which is generally lower than 0.009.

   **Line 305-308:** $P_\cup$ is highest when the return period of SH and SWH is 10-year, which is greater than 0.13 overall. $P_\cup$ is smallest when the return period for SH and SWH is 100-year, which is less than 0.015. When the return period of SH or SWH is 50-year or

100-year, the regional variation in $P_U$ are relatively small.

**Line 316-318:** Under the condition that the return period of SWH is 10-year, $P_I$ for SH with a return period of 10-year are concentrated between 0.55 and 0.75, and $P_I$ is generally less than 0.08 if the return period for SH is 100-year.

**Line 344-346:** Therefore, we calculate the change values in $P_\cap$, $P_U$, and $P_I$ for all nodes when the design return period criterion for a given variable is increased from 5-year, 10-year, 20-year, and 50-year to 10-year, 20-year, 50-year, and 100-year, respectively.

**Line 352-354:** When the return period of one variable is 10-year or 20-year, the decline in $P_\cap$ increases when the protection standard of another variable is raised. If the design criteria increase from a 50-year to a 100-year return period, the change value of $P_\cap$ decreases.

**Line 370-373:** When the protection standard for one variable is a 10-year or 20-year return period, the decrease in bivariate $P_I$ tends to increase when the design criterion for the other variable's return period is raised, but the decrease in $P_I$ is slightly reduced when the design criterion of the other variable is increased from a 50-year to a 100-year return period.

**Line 384-387:** When $RP_U$ is a 5-year return period, the design criteria of SH are between 1.5 m and 2.5 m in the eastern coastal area of the Leizhou Peninsula and fall below 0.5 m in the southeastern coastal region of Hainan Island. As the return period increases, the design surge height gradually increases, and when $RP_U$ is a 200-year return period, the design surge height in the eastern coastal area of the Leizhou Peninsula is generally higher than 3.0 m.

**Line 392-393:** When $RP_U$ is a 5-year return period, the design criteria of SWH in the coastal areas of the Leizhou Peninsula and Hainan Island are less than 2.5 m overall.

**Line 394-396:** When $RP_U$ is a 200-year return period, SWH along the coast of the Leizhou Peninsula is generally less than 6.0 m, while the design SWH along the Qiongzhou Strait and southeastern Hainan Island is relatively high.

2. **One main clarification is needed: even if the proposal is mathematically clean, the constraint condition of equation 12 based on joint probability should be discussed.**

   **A figure presenting the constraint and the objective function in the plane (x,y) could be firstly useful.**

   **Secondly, what is the physical sense of this choice? What do we practically design with this constraint condition? The authors refer to coastal protection standards. Other standards relative to coasts exist like overtoppings rate, erosion rate or wave set-up, ... Can these standards be compared with the joint probability? Do we need a**

**constraint condition on-shore or off-shore?**

**Response:** Thanks for your suggestion. The revised manuscript gives the meaning of the constraint condition in Equation 12. In addition, a schematic diagram of the constraints and the objective function is given to facilitate the reader's understanding of estimating the optimal design values for storm surge and wave combinations based on bivariate simultaneous return periods and joint return periods. The coastal protection criteria in this study only consider the SHs and SWHs, while other indicators are not currently considered and are subject to further research.

**Line 59-63:** In addition, as the intensities of the bivariates and their simultaneous probability are three-dimensional surfaces, the cross-section at a given return period is a curve rather than a specific scale value, so the joint probability of SHs and SWHs alone can not be used directly as a reference value for engineering design criteria. In order to obtain two specific scalars for SH and SWH, other constraints, such as their preferred simultaneous return periods are needed (Xu et al., 2022).

**Line 69-73:** The change in bivariate occurrence probability after increasing the engineering design criteria for the SHs and SWHs is quantitatively assessed. Finally, with the maximum bivariate simultaneous return period as the objective function and the bivariate joint return period as the constraint, the optimum engineering design values of SHs and SWHs are solved by the non-linear programming method.

**Line 420-424:** Therefore, developing appropriate design criteria for the SHs and SWHs can effectively reduce the impact of tropical cyclone marine hazards in coastal areas. Since the joint probability distribution of the bivariate is a three-dimensional surface, to obtain specific scalar values for these two hazards as design criteria, in this study, the optimal design criteria for storm surge and waves under the objective of minimum bivariate simultaneous return period are estimated using a non-linear programming approach with their estimated joint return periods as constraints.

**Line 224-241:** Based on the binary Copula function, the bivariate joint probability of extreme storm surges and waves under different joint return periods is available. In order to achieve the optimal protection effects, it is natural that we need to set the maximum bivariate simultaneous probability of SH and SWH as target functions (Eq. 16) and use joint probability as constraints (Eq. 17).

According to the non-linear programming method (Bazaraa et al., 2006), for a combined event of extreme SHs and SWHs, a series of $(x, y)$ shall be iterated to minimize $P(X > x, Y > y)$ for a given joint return period to obtain the best cost-benefit effect. Therefore, the optimal values of SH and SWH can be solved, as illustrated in Figure 4. Since we have the estimation of the joint probability for the study area instead of some specific locations, the optimal design criteria for all the eastern coasts of the Leizhou Peninsula and Hainan Island can be estimated.

[Figure]

Figure 4: Diagram of determining SH and SWH based on their joint and simultaneous return periods (red curves are joint return periods ($RP_\cup$), black curves are simultaneous return periods ($RP_\cap$), and black dots (x, y) is the optimal SH and SWH).

**Responses to Referee #2**

**General comments**

   The authors have not addressed several comments. In particular, the following answers should be revised:

   **Response:** Thanks for your suggestion. We answered some of the questions in our previous response without including all of them in the manuscript text. Following your advice, we made further explanations and added some equations to the revised manuscript. Please kindly find our response below in detail.

1.  **the criteria adopted for the generation of the unstructured grid are not clear.**

    **Response:** SMS software is used in the model to construct a refined and unstructured offshore grid of the eastern Leizhou Peninsula. A description of the unstructured grid can be found as follows:

    **Line 97-102:** The calculation region for the large-area is 105.5° E-121.2° E and 3.3° N-26.4° N, and the calculation region for the small-area is 105.5° E-116.5° E and 14.7° N-23.1° N (Figure 2a). And a gradient resolution is used to set the resolution for different regional grids. In the large-area model, the whole large-area contains 9,331 triangular grid nodes and 18,068 triangles; the resolution of the shoreline in the area near Zhanjiang is 0.07°-0.1°, while the resolution in other area is about 1 km-2 km. In the small-area model, the whole small-area contains 41,153 triangular grid nodes and 79,889 triangles; the resolution of the shoreline near Zhanjiang port is 0.0039°-0.01, the resolution of the open boundary is set to 0.1°-0.3°.

    **Line 128-130:** The calculation region for the large-area is 15° N-22° N, 110.5° E-118.5° E, which has a spatial step of 0.083° × 0.083°; the calculation region for the small-area is 21° N-21.2° N, 110° E-110.5° E, which has a spatial step of 0.0033° × 0.0033° (Figure 2b).

2.  **explain also how the boundary conditions of SWAN were set.**

    **Response:** Thank you for your suggestion. We have added the boundary conditions for the storm surge and wave models to the revised manuscript. The details are as follows:

    **Line 114-116:** The boundary condition to force the surge in the subdomain is the time series of the water level on each boundary nodes, which includes both the tide elevation of 8 major constituents (M2, S2, N2, K2, K1, O1, P1, and Q1) in that area from OSU Tidal Prediction Software and the surge elevation extracted from the full domain results (Liu et al., 2018).

**Line 130-135:** The SWAN model includes land boundaries and water boundaries, which need to be set up separately. The model assumes that the land boundary does not generate waves and assumes that the land boundary can fully absorb waves that cross or leave the shoreline. As the southern and eastern boundaries of the large-area model are open boundaries and are far from the shoreline, which is the focus of this study, the incoming wave energy at the open boundaries of the large-area model can be ignored, and the open boundary conditions for the small-area are calculated from the large-area model (Li et al., 2016).

3. **the equations should be added in the manuscript.**

**Response:** Thank you for your suggestion. We have added the governing equations for the SWAN and ADCIRC models to the revised manuscript. The equations are as follows:

**Line 135-138:** The governing equations of the SWAN model are as follows:

$$\frac{\partial N}{\partial t} + \frac{\partial}{\partial \lambda}[(c_\lambda + U)N] + cos^{-1}\varphi \frac{\partial}{\partial \varphi}[(c_\varphi + V)N\cos\varphi] + \frac{\partial}{\partial \theta}[c_\theta N] + \frac{\partial}{\partial \sigma}[c_\sigma N] = \frac{S_{tot}}{\sigma} \qquad (4)$$

The wave action density $N(t, \lambda, \varphi, \sigma, \theta)$ is allowed to evolve in time $(t)$, geographic space $(\lambda, \varphi)$ and spectral space (with relative frequencies $\sigma$ and directions $\theta$), $(c_\lambda, c_\varphi)$ is the group velocity, $(U, V)$ is the ambient current, and $c_\theta$ and $c_\sigma$ are the propagation velocities in the $\theta$- and $\sigma$- spaces, the source terms $S_{tot}$ represent wave growth by wind.

**Line 104-113:** ADCIRC computes water levels via the solution of the Generalized Wave Continuity Equation (GWCE), which is a combined and differentiated form of the continuity and momentum equations:

$$\frac{\partial^2 \zeta}{\partial t^2} + \tau_0 \frac{\partial \zeta}{\partial t} + Sp\frac{\partial \widetilde{J_\lambda}}{\partial \lambda} + \frac{\partial \widetilde{J_\varphi}}{\partial \varphi} - SpUH\frac{\partial \tau_0}{\partial \lambda} - VH\frac{\partial \tau_0}{\partial \varphi} = 0 \qquad (1)$$

and the currents are obtained from the vertically-integrated momentum equations:

$$\frac{\partial U}{\partial t} + S_p U\frac{\partial U}{\partial \lambda} + V\frac{\partial U}{\partial \varphi} - fV = -gS_p\frac{\partial}{\partial \lambda}\left[\zeta + \frac{P_s}{g\rho_0} - \alpha\eta\right] + \frac{\tau_{s\lambda,winds} + \tau_{s\lambda,winds} - \tau_{b\lambda}}{\rho_0 H} + \frac{M_\lambda - D_\lambda}{H} \qquad (2)$$

$$\frac{\partial V}{\partial t} + S_p U\frac{\partial V}{\partial \lambda} + V\frac{\partial V}{\partial \varphi} - fU = -g\frac{\partial}{\partial \varphi}\left[\zeta + \frac{P_s}{g\rho_0} - \alpha\eta\right] + \frac{\tau_{s\varphi,winds} + \tau_{s\varphi,winds} - \tau_{b\varphi}}{\rho_0 H} + \frac{M_\varphi - D_\varphi}{H} \qquad (3)$$

where $H = \zeta + h$ is total water depth; $\zeta$ is the deviation of the water surface from the mean; $h$ is bathymetric depth; $Sp = cos\varphi_0/cos\varphi$ is a spherical coordinate conversion factor and $\varphi_0$ is a reference latitude; $U$ and $V$ are depth-integrated currents in the $x-$ and $y-$ directions, respectively; $Q_\lambda = UH$ and $Q_\varphi = VH$ are fluxes per unit width; $f$ is the Coriolis parameter; $g$ is gravitational acceleration; $Ps$ is atmospheric pressure at the surface; $\rho_0$ is the reference density of water; $\eta$ is the Newtonian equilibrium tidal potential, and $\alpha$ is the effective earth elasticity factor; $\tau_{s,winds}$ and $\tau_{s,waves}$ are surface stresses due to winds and waves, respectively; $\tau_b$ is bottom stress; $M$ are lateral stress gradients; $D$ are momentum dispersion terms; and $\tau_0$ is a numerical parameter that optimizes the phase propagation properties.

**4. add a proper physical explanation for the phenomenon.**

**Response:** Thanks for your suggestion. In the revised manuscript, we have explained the significant increase in surge heights on the eastern side of Hainan Island.

**Line 262-266:** As shown in Figure 4, the SH shows a significant increasing trend as it approaches the coastline. SHs along the eastern coast of the Leizhou Peninsula are higher than most other regions. Frequent TC events, TC moving direction (Figure 1), and pocket-shaped coastal topography (Figure 2) are all favorable factors to water accumulation in this area. Another area with high SHs is located to the east of Hainan Island. Besides frequent TCs, this area is at the transition zone from the continental shelf to the continental slope, where bathymetry changes rapidly and can bring strong storm surges easily.

**5. How the dispersion phenomena were simulated offshore? Is a wave height of 30 m justifiable? This wave height was estimated precisely where the surge height shows strange values (see Figure 4).**

**Response:** Thanks for your comments. The SWHs under the specific return periods fitted by the extreme value function show an overall spatial distribution pattern of high in the southeast and low in the north. The maximum value of SWH for the 100-year event is around 20 m, and only for the return period of 200 years, the wave height is near 30m.

We also tried to explain the reasons for this:

1) The SWAN-based model simulations used in this study yielded a historical maximum wave height of about 15 m during a tropical cyclone event, which is closer to the SWH under a 100-year event than the SWH data simulated by Jian Shi and Benxia Li for coastal China. In contrast, based on SWH reanalysis data published by the ECMWF (European Centre for Medium-Range Weather Forecasts), wave reanalysis data values are significantly lower than actual observations or model simulations during tropical cyclone impacts. Therefore, in practical applications, wave model simulation data allows for more accuracy and is closer to the actual situation.

2) The 100-year and 200-year events fitted based on extreme value functions are infrequent, and the SWHs could be as high as 30m.

3) Besides the reasons mentioned above, there might be another reason. Errors could be introduced in fitting the extreme value function due to the sample size and sampling method, so there is a degree of overestimation of the effective wave height at typical recurrence periods.

**Line 269-277:** As shown in Figure 5, the SWHs near the shore are generally smaller than that in the open sea, and there is a significant decreasing trend in SWH as it gets closer to the coastline. This is mainly attributed to the shallow shore depth, island obstruction, wave breaking, and seabed friction attenuation. Among them, the SWHs in the eastern Leizhou Peninsula are lower than that of other seas, which is mainly influenced by the

curved depressed coastline and the topography of the shore section. The SWHs are influenced by the frequency, duration, and intensity of TCs, so the SWH is higher in the east and south of Hainan Island than in the north. In addition, the east side of Hainan Island from the continental shelf to the continental slope causes a wave-breaking effect and dissipation caused by the dramatic change in seafloor topography height, which results in a more significant gradient in SWH. In addition, it shall be noted that errors may be introduced during the estimation of SWHs with GEV due to the limited number of TC events as well.